# Optimizing Solution-Samplers for Combinatorial Problems: The Landscape of Policy-Gradient Methods

**Constantine Caramanis**
UT Austin & Archimedes / Athena RC
constantine@utexas.edu

**Dimitris Fotakis**
NTUA & Archimedes / Athena RC
fotakis@cs.ntua.gr

**Alkis Kalavasis**
Yale University
alvertos.kalavasis@yale.edu

**Vasilis Kontonis**
UT Austin
vkonton@gmail.com

**Christos Tzamos**
UOA & Archimedes / Athena RC
tzamos@wisc.edu

## Abstract

Deep Neural Networks and Reinforcement Learning methods have empirically shown great promise in tackling challenging combinatorial problems. In those methods a deep neural network is used as a solution generator which is then trained by gradient-based methods (e.g., policy gradient) to successively obtain better solution distributions. In this work we introduce a novel theoretical framework for analyzing the effectiveness of such methods. We ask whether there exist generative models that (i) are expressive enough to generate approximately optimal solutions; (ii) have a tractable, i.e, polynomial in the size of the input, number of parameters; (iii) their optimization landscape is benign in the sense that it does not contain sub-optimal stationary points. Our main contribution is a positive answer to this question. Our result holds for a broad class of combinatorial problems including Max- and Min-Cut, Max-$k$-CSP, Maximum-Weight-Bipartite-Matching, and the Traveling Salesman Problem. As a byproduct of our analysis we introduce a novel regularization process over vanilla gradient descent and provide theoretical and experimental evidence that it helps address vanishing-gradient issues and escape bad stationary points.

## 1 Introduction

Gradient descent has proven remarkably effective for diverse optimization problems in neural networks. From the early days of neural networks, this has motivated their use for combinatorial optimization [HT85, Smi99, VFJ15, BPL+16]. More recently, an approach by [BPL+16], where a neural network is used to generate (sample) solutions for the combinatorial problem. The parameters of the neural network thus parameterize the space of distributions. This allows one to perform gradient steps in this distribution space. In several interesting settings, including the Traveling Salesman Problem, they have shown that this approach works remarkably well. Given the widespread application but also the notorious difficulty of combinatorial optimization [GLS12, PS98, S+03, Sch05, CLS+95], approaches that provide a more general solution framework are appealing.

This is the point of departure of this paper. We investigate whether gradient descent can succeed in a general setting that encompasses the problems studied in [BPL+16]. This requires a parameterization

of distributions over solutions with a "nice" optimization landscape (intuitively, that gradient descent does not get stuck in local minima or points of vanishing gradient) and that has a polynomial number of parameters. Satisfying both requirements simultaneously is non-trivial. As we show precisely below, a simple lifting to the exponential-size probability simplex on all solutions guarantees convexity; and, on the other hand, *compressed* parameterizations with "bad" optimization landscapes are also easy to come by (we give a natural example for Max-Cut in Remark 1). Hence, we seek to understand the parametric complexity of gradient-based methods, i.e., how many parameters suffice for a benign optimization landscape in the sense that it does not contain "bad" stationary points.

We thus theoretically investigate whether there exist solution generators with a tractable number of parameters that are also efficiently optimizable, i.e., gradient descent requires a small number of steps to reach a near-optimal solution. We provide a positive answer under general assumptions and specialize our results for several classes of *hard and easy* combinatorial optimization problems, including Max-Cut and Min-Cut, Max-$k$-CSP, Maximum-Weighted-Bipartite-Matching and Traveling Salesman. We remark that a key difference between (computationally) easy and hard problems is not the ability to find a compressed and efficiently optimizable generative model but rather the ability to efficiently draw samples from the parameterized distributions.

## 1.1 Our Framework

We introduce a theoretical framework for analyzing the effectiveness of gradient-based methods on the optimization of solution generators in combinatorial optimization, inspired by [BPL+16].

Let $\mathcal{I}$ be a collection of instances of a combinatorial problem with common solution space $S$ and $L(\cdot; I) : S \to \mathbb{R}$ be the cost function associated with an instance $I \in \mathcal{I}$, i.e., $L(s; I)$ is the cost of solution $s$ given the instance $I$. For example, for the Max-Cut problem the collection of instances $\mathcal{I}$ corresponds to all graphs with $n$ nodes, the solution space $S$ consists of all subsets of nodes, and the loss $L(s; I)$ is equal to (minus) the weight of the cut $(s, [n] \setminus s)$ corresponding to the subset of nodes $s \in S$ (our goal is to minimize $L$).

**Definition 1** (Solution Cost Oracle). *For a given instance $I$ we assume that we have access to an oracle $\mathcal{O}(\cdot; I)$ to the cost of any given solution $s \in S$, i.e., $\mathcal{O}(s; I) = L(s; I)$.*

The above oracle is standard in combinatorial optimization and query-efficient algorithms are provided for various problems [RSW17, GPRW19, LSZ21, AEG+22, PRW22]. We remark that the goal of this work is not to design algorithms that solve combinatorial problems using access to the solution cost oracle (as the aforementioned works do). This paper focuses on landscape design: the algorithm is **fixed**, namely (stochastic) gradient descent; the question is how to design a generative model that has a small number of parameters and the induced optimization landscape allows gradient-based methods to converge to the optimal solution without getting trapped at local minima or vanishing gradient points.

Let $\mathcal{R}$ be some prior distribution over the instance space $\mathcal{I}$ and $\mathcal{W}$ be the space of parameters of the model. We now define the class of solution generators. The solution generator $p(w)$ with **parameter** $w \in \mathcal{W}$ takes as **input** an instance $I$ and **generates a random solution** $s$ in $S$. To distinguish between the output, the input, and the parameter of the solution generator, we use the notation $p(\cdot; I; w)$ to denote the distribution over solutions and $p(s; I; w)$ to denote the probability of an individual solution $s \in S$. We denote by $\mathcal{P} = \{p(w) : w \in \mathcal{W}\}$ the above parametric class of solution generators. For some parameter $w$, the loss corresponding to the solutions sampled by $p(\cdot; I; w)$ is equal to

$$\mathcal{L}(w) = \mathop{\mathbf{E}}_{I \sim \mathcal{R}}[\mathcal{L}_I(w)], \quad \mathcal{L}_I(w) = \mathop{\mathbf{E}}_{s \sim p(\cdot; I; w)}[L(s; I)]. \tag{1}$$

Our goal is to optimize the parameter $w \in \mathcal{W}$ in order to find a sampler $p(\cdot; I; w)$ whose loss $\mathcal{L}(w)$ is close to the expected optimal value opt:

$$\text{opt} = \mathop{\mathbf{E}}_{I \sim \mathcal{R}} \left[ \min_{s \in S} L(s; I) \right]. \tag{2}$$

The policy gradient method [Kak01] expresses the gradient of $\mathcal{L}$ as follows

$$\nabla_w \mathcal{L}(w) = \mathop{\mathbf{E}}_{I \sim \mathcal{R}} \mathop{\mathbf{E}}_{s \sim p(\cdot; I; w)}[L(s; I) \nabla_w \log p(s; I; w)],$$

and updates the parameter $w$ using the gradient descent update. Observe that a (stochastic) policy gradient update can be implemented using only access to a solution cost oracle of Definition 1.

**Solution Generators.** In [BPL+16] the authors used neural networks as parametric solution generators for the TSP problem. They provided empirical evidence that optimizing the parameters of the neural network using the policy gradient method results to samplers that generate very good solutions for (Euclidean) TSP instances. Parameterizing the solution generators using neural networks essentially *compresses* the description of distributions over solutions (the full parameterization would require assigning a parameter to every solution-instance pair $(s, I)$). Since for most combinatorial problems the size of the solution space is exponentially large (compared to the description of the instance), it is crucial that for such methods to succeed the parameterization must be *compressed* in the sense that the description of the parameter space $\mathcal{W}$ is polynomial in the size of the description of the instance family $\mathcal{I}$. Apart from having a tractable number of parameters, it is important that the *optimization objective* corresponding to the parametric class $\mathcal{P}$ can provably be optimized using some first-order method in polynomial (in the size of the input) iterations.

We collect these desiderata in the following definition. We denote by $[\mathcal{I}]$ the description size of $\mathcal{I}$, i.e., the number of bits required to identify any element of $\mathcal{I}$. For instance, if $\mathcal{I}$ is the space of unweighted graphs with at most $n$ nodes, $[\mathcal{I}] = O(n^2)$.

**Definition 2** (Complete, Compressed and Efficiently Optimizable Solution Generator). *Fix a prior $\mathcal{R}$ over $\mathcal{I}$, a family of solution generators $\mathcal{P} = \{p(w) : w \in \mathcal{W}\}$, a loss function $\mathcal{L}$ as in Equation (1) and some $\epsilon > 0$.*

1. *We say that $\mathcal{P}$ is **complete** if there exists some $\overline{w} \in \mathcal{W}$ such that $\mathcal{L}(\overline{w}) \leq \mathrm{opt} + \varepsilon$, where $\mathrm{opt}$ is defined in (2).*

2. *We say that $\mathcal{P}$ is **compressed** if the description size of the parameter space $\mathcal{W}$ is polynomial in $[\mathcal{I}]$ and in $\log(1/\varepsilon)$.*

3. *We say that $\mathcal{P}$ is **efficiently optimizable** if there exists a first-order method applied on the objective $\mathcal{L}$ such that after $T = \mathrm{poly}([\mathcal{W}], 1/\varepsilon)$ many updates on the parameter vectors, finds an (at most) $\epsilon$-sub-optimal vector $\widehat{w}$, i.e., $\mathcal{L}(\widehat{w}) \leq \mathcal{L}(\overline{w}) + \epsilon$.*

**Remark 1.** *We remark that constructing parametric families that are complete and compressed, complete and efficiently optimizable, or compressed and efficiently optimizable (i.e., satisfying any pair of assumptions of Question 1 but not all 3) is usually a much easier task. Consider, for example, the Max-Cut problem on a fixed (unweighted) graph with $n$ nodes. Note that $\mathcal{I}$ has description size $O(n^2)$. Solutions of the Max-Cut for a graph with $n$ nodes are represented by vertices on the binary hypercube $\{\pm 1\}^n$ (coordinate $i$ dictates the side of the cut that we put node $i$). One may consider the full parameterization of all distributions over the hypercube. It is not hard to see that this is a **complete and efficiently optimizable** family (the optimization landscape corresponds to optimizing a linear objective). However, it **is not compressed**, since it requires $2^n$ parameters. On the other extreme, considering a product distribution over coordinates, i.e., we set the value of node $i$ to be $+1$ with probability $p_i$ and $-1$ with $1-p_i$ gives a **complete and compressed** family. However, as we show in Appendix B, the landscape of this compressed parameterization suffers from highly sub-optimal local minima and therefore, it is **not efficiently optimizable**.*

Therefore, in this work we investigate whether it is possible to have all 3 desiderata of Definition 2 *at the same time.*

**Question 1.** *Are there complete, compressed, and efficiently optimizable solution generators (i.e., satisfying Definition 2) for challenging combinatorial tasks?*

## 1.2 Our Results

**Our Contributions.** Before we present our results formally, we summarize the contributions of this work.

- Our main contribution is a positive answer (Theorem 1) to Question 1 under general assumptions that capture many combinatorial tasks. We identify a set of conditions (see Assumption 1) that allow us to design a family of solution generators that are complete, compressed and efficiently optimizable.
- The conditions are motivated by obstacles that are important for any approach of this nature. This includes solutions that escape to infinity, and also parts of the landscape with vanishing gradient. See the discussion in Section 3 and Figure 1.

- We specialize our framework to several important combinatorial problems, some of which are NP-hard, and others tractable: Max-Cut, Min-Cut, Max-$k$-CSP, Maximum-Weight-Bipartite-Matching, and the Traveling Salesman Problem.

- Finally, we investigate experimentally the effect of the entropy regularizer and the fast/slow mixture scheme that we introduced (see Section 3) and provide evidence that it leads to better solution generators.

We begin with the formal presentation of our assumptions on the feature mappings of the instances and solutions and on the structure of cost function of the combinatorial problem.

**Assumption 1** (Structured Feature Mappings). *Let $S$ be the solution space and $\mathcal{I}$ be the instance space. There exist feature mappings $\psi_S : S \to X$, for the solutions, and, $\psi_{\mathcal{I}} : \mathcal{I} \to Z$, for the instances, where $X, Z$ are Euclidean vector spaces of dimension $n_X$ and $n_Z$, such that*

1. *(Bounded Feature Spaces) The feature and instance mappings are bounded, i.e., there exist $D_S, D_{\mathcal{I}} > 0$ such that $\|\psi_S(s)\|_2 \leq D_S$, for all $s \in S$ and $\|\psi_{\mathcal{I}}(I)\|_2 \leq D_{\mathcal{I}}$, for all $I \in \mathcal{I}$.*

2. *(Bilinear Cost Oracle) The cost of a solution $s$ under instance $I$ can be expressed as a bilinear function of the corresponding feature vector $\psi_S(s)$ and instance vector $\psi_{\mathcal{I}}(I)$, i.e., the solution oracle can be expressed as $\mathcal{O}(s, I) = \psi_{\mathcal{I}}(I)^\top M \psi_S(s)$ for any $s \in S, I \in \mathcal{I}$, for some matrix $M$ with $\|M\|_{\mathrm{F}} \leq C$.*

3. *(Variance Preserving Features) There exists $\alpha > 0$ such that $\mathbf{Var}_{s \sim U(S)}[v \cdot \psi_S(s)] \geq \alpha \|v\|_2^2$ for any $v \in X$, where $U(S)$ is the uniform distribution over the solution space $S$.*

4. *(Bounded Dimensions/Diameters) The feature dimensions $n_X, n_Z$, and the diameter bounds $D_S, D_{\mathcal{I}}, C$ are bounded above by a polynomial of the description size of the instance space $\mathcal{I}$. The variance lower bound $\alpha$ is bounded below by $1/\mathrm{poly}([\mathcal{I}])$.*

**Remark 2** (Boundedness and Bilinear Cost Assumptions). *We remark that Items 1, 4 are simply boundedness assumptions for the corresponding feauture mappings and usually follow easily assuming that we consider reasonable feature mappings. At a high-level, the assumption that the solution is a bilinear function of the solution and instance features (Item 2) prescribes that "good" feature mappings should enable a simple (i.e., bilinear) expression for the cost function. In the sequel we see that this is satisfied by natural feature mappings for important classes of combinatorial problems.*

**Remark 3** (Variance Preservation Assumption). *In Item 3 (variance preservation) we require that the solution feature mapping has variance along every direction, i.e., the feature vectors corresponding to the solutions must be "spread-out" when the underlying solution generator is the uniform distribution. As we show, this assumption is crucial so that the gradients of the resulting optimization objective are not-vanishing, allowing for its efficient optimization.*

We mention that various important combinatorial problems satisfy Assumption 1. For instance, Assumption 1 is satisfied by Max-Cut, Min-Cut, Max-$k$-CSP, Maximum-Weight-Bipartite-Matching and Traveling Salesman. We refer the reader to the upcoming Section 2 for an explicit description of the structured feature mappings for these problems. Having discussed Assumption 1, we are ready to state our main abstract result which resolves Question 1.

**Theorem 1.** *Consider a combinatorial problem with instance space $\mathcal{I}$ that satisfies Assumption 1. For any prior $\mathcal{R}$ over $\mathcal{I}$ and $\epsilon > 0$, there exists a family of solution generators $\mathcal{P} = \{p(w) : w \in \mathcal{W}\}$ with parameter space $\mathcal{W}$ that is complete, compressed and, efficiently optimizable.*

A sketch behind the design of the family $\mathcal{P}$ can be found in Section 3 and Section 4.

**Remark 4** (Computational Barriers in Sampling). *We note that the families of generative models (a.k.a., solution generators) that we provide have polynomial parameter complexity and are optimizable in a small number of steps using gradient-based methods. Hence, in a small number of iterations, gradient-based methods converge to distributions whose mass is concentrated on nearly optimal solutions. This holds, as we show, even for challenging (NP-hard) combinatorial problems. Our results do not, however, prove $\mathrm{P} = \mathrm{NP}$, as it may be computationally hard to sample from our generative models. We remark that while such approaches are in theory hard, such models seem to perform remarkably well experimentally where sampling is based on Langevin dynamics techniques [SE20, SSDK+20]. Though as our theory predicts, and simulations support, landscape problems seem to be a direct impediment even to obtain good approximate solutions.*

**Remark 5** (Neural Networks as Solution Samplers). *A natural question would be whether our results can be extended to the case where neural networks are (efficient) solution samplers, as in [BPL+16]. Unfortunately, a benign landscape result for neural network solution generators most likely cannot exist. It is well-known that end-to-end theoretical guarantees for training neural networks are out of reach since the corresponding optimization tasks are provably computationally intractable, see, e.g., [CGKM22] and the references therein.*

Finally, we would like to mention an interesting aspect of Assumption 1. Given a combinatorial problem, Assumption 1 essentially asks for the *design* of feature mappings for the solutions and the instances that satisfy desiderata such as boundedness and variance preservation. Max-Cut, Min-Cut, TSP and Max-$k$-CSP and other problems satisfy Assumption 1 because we managed to design appropriate (problem-specific) feature mappings that satisfy the requirements of Assumption 1. There are interesting combinatorial problems for which we do not know how to design such good feature mappings. For instance, the "natural" feature mapping for the Satisfiability problem (SAT) (similar to the one we used for Max-$k$-CSPs) would require feature dimension exponential in the size of the instance (we need all possible monomials of $n$ variables and degree at most $n$) and therefore, would violate Item 4 of Assumption 1.

## 1.3 Related Work

**Neural Combinatorial Optimization.** Tackling combinatorial optimization problems constitutes one of the most fundamental tasks of theoretical computer science [GLS12, PS98, S+03, Sch05, CLS+95] and various approaches have been studied for these problems such as local search methods, branch-and-bound algorithms and meta-heuristics such as genetic algorithms and simulated annealing. Starting from the seminal work of [HT85], researchers apply neural networks [Smi99, VFJ15, BPL+16] to solve combinatorial optimization tasks. In particular, researchers have explored the power of machine learning, reinforcement learning and deep learning methods for solving combinatorial optimization problems [BPL+16, YW20, LZ09, DCL+18, BLP21, MSIB21, NOST18, SHM+16, MKS+13, SSS+17, ER18, KVHW18, ZCH+20, CCK+21, MGH+19, GCF+19, KLMS19].

The use of neural networks in combinatorial problems is extensive [SLB+18, JLB19, GCF+19, YGS20, MSIB21, BPL+16, KDZ+17, YP19, CT19, YBV19, KCK+20, KCY+21, DAT20, NJS+20, TRWG21, AMW18, KL20, Jeg22, SBK22, ART23] and various papers aim to understand the theoretical ability of neural networks to solve such problems [HS23b, HS23a, Gam23]. Our paper builds on the framework of the influential experimental work of [BPL+16] to tackle combinatorial optimization problems such as TSP using neural networks and reinforcement learning. [KP+21] uses an entropy maximization scheme in order to generate diversified candidate solutions. This experimental heuristic is quite close to our theoretical idea for entropy regularization. In our work, entropy regularization allows us to design quasar-convex landscapes and the fast/slow mixing scheme to obtain diversification of solutions. Among other related applied works, [KCK+20, KPP22] study the use of Transformer architectures combined with the Reinforce algorithm employing symmetries (i.e., the existence of multiple optimal solutions of a CO problem) improving the generalization capability of Deep RL NCO and [MLC+21] studies Transformer architectures and aims to learn improvement heuristics for routing problems using RL.

**Gradient Descent Dynamics.** Our work provides theoretical understanding on the gradient-descent landscape arising in NCO problems. Similar questions regarding the dynamics of gradient descent have been studied in prior work concerning neural networks; for instance, [AS20] and [AKM+21] fix the algorithm (SGD on neural networks) and aim to understand the power of this approach (which function classes can be learned). Various other works study gradient descent dynamics in neural networks. We refer to [AS18, AS20, ABAB+21, MYSSS21, BEG+22, DLS22, ABA22, AAM22, BBSS22, ABAM23, AKM+21, EGK+23] (and the references therein) for a small sample of this line of research.

## 2 Combinatorial Applications

We now consider concrete combinatorial problems and show that there exist appropriate and natural feature mappings for the solutions and instances that satisfy Assumption 1; so Theorem 1 is applicable for any such combinatorial task. For a more detailed treatment, we refer to Appendix G.

**Min-Cut and Max-Cut.** Min-Cut (resp. Max-Cut) are central graph combinatorial problems where the task is to split the nodes of the graph in two subsets so that the number of edges from one subset to the other (edges of the cut) is minimized (resp. maximized). Given a graph $G$ with $n$ nodes represented by its Laplacian matrix $L_G = D - A$, where $D$ is the diagonal degree matrix and $A$ is the adjacency matrix of the graph, the goal in the Min-Cut (resp. Max-Cut) problem is to find a solution vector $s \in \{\pm 1\}^n$ so that $s^\top L_G s / 4$ is minimized (resp. maximized).

We first show that there exist natural feature mappings so that the cost of every solution $s$ under any instance/graph $G$ is a bilinear function of the feature vectors, see Item 2 of Assumption 1. We consider the correlation-based feature mapping $\psi_S(s) = (ss^\top)^\flat \in \mathbb{R}^{n^2}$, where by $(\cdot)^\flat$ we denote the vectorization/flattening operation and the Laplacian for the instance (graph), $\psi_\mathcal{I}(G) = (L_G)^\flat \in \mathbb{R}^{n^2}$. Then simply setting the matrix $M$ to be the identity $I \in \mathbb{R}^{n^2 \times n^2}$ the cost of any solution $s$ can be expressed as the bilinear function $\psi_\mathcal{I}(G)^\top M \psi_S(s) = (L_G^\flat)^\top (ss^\top)^\flat = s^\top L_G s$. We observe that for (unweighted) graphs with $n$ nodes the description size of the family of all instances $\mathcal{I}$ is roughly $O(n^2)$, and therefore the dimensions of the feature mappings $\psi_S, \psi_\mathcal{I}$ are clearly polynomial in the description size of $\mathcal{I}$. Moreover, considering unweighted graphs, it holds that $\|\psi_\mathcal{I}(G)\|_2, \|\psi_S(s)\|_2, \|M\|_F \leq \mathrm{poly}(n)$. Therefore, the constants $D_S, D_\mathcal{I}, C$ are polynomial in the description size of the instance family.

It remains to show that our solution feature mapping satisfies the variance preservation assumption, i.e., Item 3 in Assumption 1. A uniformly random solution vector $s \in \{\pm 1\}^n$ is sampled by setting each $s_i = 1$ with probability $1/2$ independently. In that case, we have $\mathbf{E}[v \cdot x] = 0$ and therefore $\mathbf{Var}(v \cdot x) = \mathbf{E}[(v \cdot x)^2] = \sum_{i \neq j} v_i v_j \mathbf{E}[x_i x_j] = \sum_i v_i^2 = \|v\|_2^2$, since, by the independence of $x_i, x_j$, the cross-terms of the sum vanish. We observe that the same hold true for the Max-Cut problem and therefore, structured feature mappings exist for Max-Cut as well (where $L(s; G) = -s^\top L_G s$). We shortly mention that there also exist structured feature mappings for Max-$k$-CSP. We refer to Theorem 4 for further details.

**Remark 6** (Partial Instance Information/Instance Context). *We remark that Assumption 1 allows for the "instance" $I$ to only contain partial information about the actual cost function. For example, consider the setting where each sampled instance is an unweighted graph $G$ but the cost oracle takes the form $\mathcal{O}(G, s) = (L_G)^\flat M (ss^\top)^\flat$ for a matrix $M_{ij} = a_i$ when $i = j$ and $M_{ij} = 0$ otherwise. This cost function models having a **unknown weight function**, i.e., the weight of edge $i$ of $G$ is $a_i$ if edge $i$ exists in the observed instance $G$, on the edges of the observed unweighted graph $G$, that the algorithm has to learn in order to be able to find the minimum or maximum cut. For simplicity, in what follows, we will continue referring to $I$ as the instance even though it may only contain partial information about the cost function of the underlying combinatorial problem.*

**Maximum-Weight-Bipartite-Matching and TSP.** The Maximum-Weight-Bipartite-Matching (MWBP) problem is another graph problem that, given a bipartite graph $G$ with $n$ nodes and $m$ edges, asks for the maximum-weight matching. The feature vector corresponding to a matching can be represented as a binary matrix $R \in \{0, 1\}^{n \times n}$ with $\sum_j R_{ij} = 1$ for all $i$ and $\sum_i R_{ij} = 1$ for all $j$, i.e., $R$ is a permutation matrix. Therefore, for a candidate matching $s$, we set $\psi_S(s)$ to be the matrix $R$ defined above. Moreover, the feature vector of the graph is the (negative flattened) adjacency matrix $E^\flat$. The cost oracle is then $L(R; E) = \sum_{ij} E_{ij} M_{ij} R_{ij}$ perhaps for an unknown weight matrix $M_{ij}$ (see Remark 6). For the Traveling Salesman Problem (TSP) the feature vector is again a matrix $R$ with the additional constraint that $R$ has to represent a single cycle (a tour over all cities). The cost function for TSP is again $L(R; E) = \sum_{ij} E_{ij} M_{ij} R_{ij}$. One can check that those representations of the instance and solution satisfy the assumptions of Items 1 and 4. Showing that the variance of those representations has a polynomial lower bound is more subtle and we refer the reader to the Supplementary Material.

We shortly mention that the solution generators for Min-Cut and Maximum-Weight-Bipartite-Matching are also efficiently samplable.

## 3 Optimization Landscape

**Exponential Families as Solution Generators.** A natural candidate to construct our family of solution generators is to consider the distribution that assigns to each solution $s \in S$ and instance $I \in \mathcal{I}$ mass proportional to its score $\exp(-\tau L(s; I)) = \exp(-\tau \psi_\mathcal{I}(I)^\top M \psi_S(s)) = \exp(-\tau z^\top M x)$

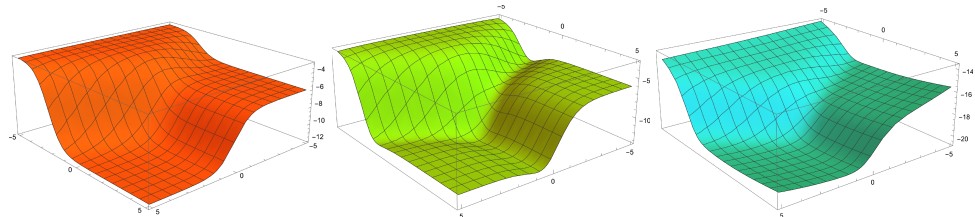

Figure 1: In the **left** plot, we show the landscape of the "vanilla" objective of Eq.(1) for the feature domain $X = \{(1,0),(2,2),(0,2)\}$ and linear cost oracle $c \cdot x$ for $c = (-3,-3)$. We see that the "vanilla" objective is minimized at the direction of $-c$, i.e., along the direction $\tau(1,1)$ for $\tau \to +\infty$. We observe the two issues described in Section 3, i.e., that the true minimizer is a point at infinity, and that gradients vanish so gradient descent may get trapped in sub-optimal solutions, (e.g., in the upper-right corner if initialized in the top corner). In the **middle** plot, we show the landscape of the entropy-regularized objective of Eq.(3) that makes the minimizer finite and brings it closer to the origin. Observe that even if a gradient iteration is initialized in the top corner it will eventually converge to the minimizer; however the rate of convergence may be very slow. The **right** plot corresponds to the loss objective where we combine a mixture of exponential families as solution generator, as in Eq.(5), and the entropy regularization approach. We observe that we are able to obtain a benign (quasar-convex) landscape via the entropy regularization while the mixture-generator guarantees non-vanishing gradients.

for some "temperature" parameter $\tau$, where $\psi_{\mathcal{I}}$ and $\psi_S$ are the feature mappings promised to exist due to Assumption 1, $z = \psi_{\mathcal{I}}(I)$, and, $x = \psi_S(s)$. Note that as long as $\tau \to +\infty$, this distribution tends to concentrate on solutions that achieve small loss.

**Remark 7.** *To construct the above solution sampler one could artificially query specific solutions to the cost oracle of Definition 1 and try to learn the cost matrix $M$. However, we remark that our goal (see Definition 2) is to show that we can train a parametric family via gradient-based methods so that it generates (approximately) optimal solutions and not to simply learn the cost matrix $M$ via some other method and then use it to generate good solutions.*

**Obstacle I: Minimizers at Infinity.** One could naturally consider the parametric family $\phi(x; z; W) \propto \exp(z^\top W x)$ (note that with $W = -\tau M$, we recover the distribution of the previous paragraph) and try to perform gradient-based methods on the loss (recall that $L(x; z) = z^\top M x$)[1]

$$\mathcal{L}(W) = \mathop{\mathbf{E}}_{z \sim \mathcal{R}} \mathop{\mathbf{E}}_{x \sim \phi(\cdot; z; W)} \left[ z^\top M x \right]. \tag{1}$$

The question is whether gradient updates on the parameter $W$ eventually converge to a matrix $\overline{W}$ whose associated distribution $\phi(\overline{W})$ generates near-optimal solutions (note that the matrix $-\tau M$ with $\tau \to +\infty$ is such a solution). After computing the gradient of $\mathcal{L}$, we observe that

$$\nabla_W \mathcal{L}(W) \cdot M = \mathbf{Var}_{z \sim \mathcal{R}, x \sim \phi(\cdot; z; W)}[z^\top M x] \geq 0 \,,$$

where the inner product between two matrices $A \cdot B$ is the trace $\mathrm{Tr}(A^\top B) = \sum_{i,j} A_{ij} B_{ij}$. This means that the gradient field of $\mathcal{L}$ always has a contribution to the direction of $M$. Nevertheless the actual minimizer is at infinity, i.e., it corresponds to the point $\overline{W} = -\tau M$ when $\tau \to +\infty$. While the correlation with the optimal point is positive (which is encouraging), having such contribution to this direction is not a sufficient condition for actually reaching $\overline{W}$. The objective has vanishing gradients at infinity and gradient descent may get trapped in sub-optimal stationary points, see the left plot in Figure 1.

**Solution I: Quasar Convexity via Entropy Regularization.** Our plan is to try and make the objective landscape more benign by adding an entropy-regularizer. Instead of trying to make the objective convex (which may be too much to ask in the first place) *we are able obtain a much better landscape with a finite global minimizer and a gradient field that guides gradient descent to the minimizer.* Those properties are described by the so-called class of "quasar-convex" functions. Quasar convexity (or weak quasi-convexity [HMR16]) is a well-studied notion in optimization [HMR16, HSS20, LV16, ZMB$^+$17, HLSS15] and can be considered as a high-dimensional generalization of unimodality.

---

[1]We note that we overload the notation and assume that our distributions generate directly the featurizations $z$ (resp. $x$) of $I$ (resp. $s$).

**Definition 3** (Quasar Convexity [HMR16, HSS20])**.** *Let $\gamma \in (0, 1]$ and let $\overline{x}$ be a minimizer of the differentiable function $f : \mathbb{R}^n \to \mathbb{R}$. The function $f$ is $\gamma$-**quasar-convex** with respect to $\overline{x}$ on a domain $D \subseteq \mathbb{R}^n$ if for all $x \in D$, $\nabla f(x) \cdot (x - \overline{x}) \geq \gamma(f(x) - f(\overline{x}))$.*

In the above definition, notice that the main property that we need to establish is that the gradient field of our objective correlates positively with the direction $W - \overline{W}$, where $\overline{W}$ is its minimizer. We denote by $H : \mathcal{W} \to \mathbb{R}$ the negative entropy of $\phi(W)$, i.e.,

$$H(W) = \mathop{\mathbf{E}}_{z \sim \mathcal{R}} \mathop{\mathbf{E}}_{x \sim \phi(\cdot;z;W)} [\log \phi(x; z; W)], \tag{2}$$

and consider the *regularized* objective

$$\mathcal{L}_\lambda(W) = \mathcal{L}(W) + \lambda H(W), \tag{3}$$

for some $\lambda > 0$. We show (follows from Lemma 4) that the gradient-field of the regularized objective indeed "points" towards a finite minimizer (the matrix $\overline{W} = -M/\lambda$):

$$\nabla_W \mathcal{L}_\lambda(W) \cdot (W + M/\lambda) =$$
$$\mathbf{Var}[z^\top (W + M/\lambda)x] \geq 0, \tag{4}$$

where the randomness is over $z \sim \mathcal{R}, x \sim \phi(\cdot; z; W)$. Observe that now the minimizer of $\mathcal{L}_\lambda$ is the point $-M/\lambda$, which for $\lambda = \text{poly}(\epsilon, \alpha, 1/C, 1/D_S, 1/D_\mathcal{I})$ (these are the parameters of Assumption 1) is promised to yield a solution sampler that generates $\epsilon$-sub-optimal solutions (see also Proposition 2 and Appendix C). Having the property of Equation (4) suffices for showing that a gradient descent iteration (with an appropriately small step-size) will *eventually* converge to the minimizer.

**Obstacle II: Vanishing Gradients.** While we have established that the gradient field of the regularized objective "points" towards the right direction, the regularized objective still suffers from vanishing gradients, see the middle plot in Figure 1. In other words, $\gamma$ in the definition of quasar convexity (Definition 3) may be exponentially small, as it is proportional to the variance of the random variable $z^\top (W + M/\lambda)x$, see Equation (4). As we see in the middle plot of Figure 1, the main issue is the vanishing gradient when $W$ gets closer to the minimizer $-M/\lambda$ (towards the front-corner). For simplicity, consider the variance along the direction of $W$, i.e., $\mathbf{Var}[z^\top W x]$ and recall that $x$ is generated by the density $\exp(z^\top W x)/(\sum_{x \in X} \exp(z^\top W x))$. When $\|W\|_2 \to +\infty$ we observe that the value $z^\top W x$ concentrates exponentially fast to $\max_{x \in X} z^\top W x$ (think of the convergence of the soft-max to the max function). Therefore, the variance $\mathbf{Var}[z^\top W x]$ may vanish exponentially fast making the convergence of gradient descent slow.

**Solution II: Non-Vanishing Gradients via Fast/Slow Mixture Generators.** We propose a fix to the vanishing gradients issue by using a mixture of exponential families as a solution generator. We define the family of solution generators $\mathcal{P} = \{p(W) : W \in \mathcal{W}\}$ to be

$$\mathcal{P} = \{(1 - \beta^\star)\phi(W) + \beta^\star \phi(\rho^\star W) : W \in \mathcal{W}\}, \tag{5}$$

for a (fixed) mixing parameter $\beta^\star$ and a (fixed) temperature parameter $\rho^\star$. The main idea is to have the first component of the mixture to converge fast to the optimal solution (to $-M/\lambda$) while the second "slow" component that has parameter $\rho^\star W$ stays closer to the uniform distribution over solutions that guarantees non-trivial variance (and therefore non-vanishing gradients).

More precisely, taking $\rho^\star$ to be sufficiently small, the distribution $\phi(\rho^\star W)$ is *almost uniform* over the solution space $\psi_S(S)$. Therefore, in Equation (4), the almost uniform distribution component of the mixture will add to the variance and allow us to show a lower bound. This is where Item 3 of Assumption 1 comes into play and gives us the desired non-trivial variance lower bound under the uniform distribution. We view this fast/slow mixture technique as an interesting insight of our work: we use the "fast" component (the one with parameter $W$) to actually reach the optimal solution $-M/\lambda$ and and we use the "slow" component (the one with parameter $\rho^\star W$ that essentially generates random solutions) to preserve a non-trivial variance lower bound during optimization.

## 4 Complete, Compressed and Efficiently Optimizable Solution Generators

In this section, we discuss the main results that imply Theorem 1: the family $\mathcal{P}$ of Equation (5) is complete, compressed and efficiently optimizable (for some choice of $\beta^\star, \rho^\star$ and $\mathcal{W}$).

**Completeness.** First, we show that the family of solution generators of Equation (5) is complete. For the proof, we refer to Proposition 2 in Appendix C. At a high-level, we to pick $\beta^\star, \rho^\star$ to be of order poly$(\epsilon, \alpha, 1/C, 1/D_S, 1/D_\mathcal{I})$. This yields that the matrix $\overline{W} = -M/\lambda$ is such that $\mathcal{L}(\overline{W}) \leq \mathrm{opt} + \epsilon$, where $M$ is the matrix of Item 2 in Assumption 1 and $\lambda$ is poly$(\epsilon/[\mathcal{I}])$. To give some intuition about this choice of matrix, let us see how $\mathcal{L}(\overline{W})$ behaves. By definition, we have that

$$\mathcal{L}(\overline{W}) = \mathop{\mathbf{E}}_{z \sim \mathcal{R}} \mathop{\mathbf{E}}_{x \sim p(\cdot; z; \overline{W})} \left[ z^\top M x \right],$$

where the distribution $p$ belongs to the family of Equation (5), i.e., $p(\overline{W}) = (1 - \beta^\star)\phi(\overline{W}) + \beta^\star\phi(\rho^\star W)$. Since the mixing weight $\beta^\star$ is small, we have that $p(\overline{W})$ is approximately equal to $\phi(\overline{W})$. This means that our solution generator draws samples from the distribution whose mass at $x$ given instance $z$ is proportional to $\exp(-z^\top M x/\lambda)$ and, since $\lambda > 0$ is very small, the distribution concentrates to solutions $x$ that tend to minimize the objective $z^\top M x$. This is the reason why $\overline{W} = -M/\lambda$ is close to opt in the sense that $\mathcal{L}(\overline{W}) \leq \mathrm{opt} + \epsilon$.

**Compression.** As a second step, we show (in Proposition 3, see Appendix D) that $\mathcal{P}$ is a compressed family of solution generators. This result follows immediately from the structure of Equation (5) (observe that $W$ has $n_X \, n_Z$ parameters) and the boundedness of $\overline{W} = -M/\lambda$.

**Efficiently Optimizable.** The proof of this result essentially corresponds to the discussion provided in Section 3. Our main structural result shows that the landscape of the regularized objective with the fast/slow mixture solution-generator is quasar convex. More precisely, we consider the following objective:

$$\mathcal{L}_\lambda(W) = \mathop{\mathbf{E}}_{z \sim \mathcal{R}} \mathop{\mathbf{E}}_{x \sim p(\cdot; z; W)} [z^\top M x] + \lambda R(W), \tag{1}$$

where $p(W)$ belongs in the family $\mathcal{P}$ of Equation (5) and $R$ is a weighted sum of two negative entropy regularizers (to be in accordance with the mixture structure of $\mathcal{P}$), i.e., $R(W) = (1 - \beta^\star)H(W) + \beta^\star/\rho^\star H(\rho^\star W)$. Our main structural results follows (for the proof, see Appendix E.1).

**Proposition 1** (Quasar Convexity). *Consider $\epsilon > 0$ and a prior $\mathcal{R}$ over $\mathcal{I}$. Assume that Assumption 1 holds. The function $\mathcal{L}_\lambda$ of Equation (1) with domain $\mathcal{W}$ is* poly$(\epsilon, \alpha, 1/C, 1/D_S, 1/D_\mathcal{I})$-*quasar convex with respect to $-M/\lambda$ on the domain $\mathcal{W}$.*

Since $\rho^\star$ is small (by Proposition 2), $H(\rho^\star W)$ is essentially constant and close in value to the negative entropy of the uniform distribution. Hence, the effect of $R(W)$ during optimization is essentially the same as that of $H(W)$ (since $\beta^\star$ is close to 0). We show that $\mathcal{L}_\lambda$ is quasar convex with a non-trivial parameter $\gamma$ (see Proposition 1). We can then apply (in a black-box manner) the convergence results from [HMR16] to optimize it using projected SGD. We show that SGD finds a weight matrix $\widehat{W}$ such that the solution generator $p(\widehat{W})$ generates solutions achieving actual loss $\mathcal{L}$ close to that of the near optimal matrix $\overline{W} = -M/\lambda$, i.e., $\mathcal{L}(\widehat{W}) \leq \mathcal{L}(\overline{W}) + \epsilon$. For further details, see Appendix E.3.

# 5 Experimental Evaluation

In this section, we investigate experimentally the effect of our main theoretical contributions, the entropy regularizer (see Equation (2)) and the fast/slow mixture scheme (see Equation (5)). We try to find the Max-Cut of a fixed graph $G$, i.e., the support of the prior $\mathcal{R}$ is a single graph. Similarly to our theoretical results, our sampler is of the form $e^{\mathrm{score}(s; w)}$, where $s \in \{-1, 1\}^n$ (here $n$ is the number of nodes in the graph) is a candidate solution of the Max-Cut problem. For the score function we used a simple linear layer (left plot of Figure 2) and a 3-layer ReLU network (right plot of Figure 2).

**Small Graph Instances.** Focusing on instances where the number of nodes $n$ is small (say $n = 15$), we can explicitly compute the density function and work with an *exact* sampler. We generate 100 random $G(n, p)$ graphs with $n = 15$ nodes and $p = 0.5$ and train solution generators using both the "vanilla" loss $\mathcal{L}$ and the entropy-regularized loss $\mathcal{L}_\lambda$ with the fast/slow mixture scheme. We perform 600 iterations and, for the entropy regularization, we progressively decrease the regularization weight, starting from 10, and dividing it by 2 every 60 iterations. Out of the 100 trials we found that our proposed objective was always able to find the optimal cut while the model trained with the vanilla loss was able to find it for approximately $65\%$ of the graphs (for 65 out of 100 using the linear network and for 66 using the ReLU network).

Hence, our experiments demonstrate that while the unregularized objective is often "stuck" at sub-optimal solutions – and this happens even for very small instances ($n =15$ nodes) – of the Max-Cut problem, the objective motivated by our theoretical results is able to find the optimal solutions. For further details, see Appendix I.

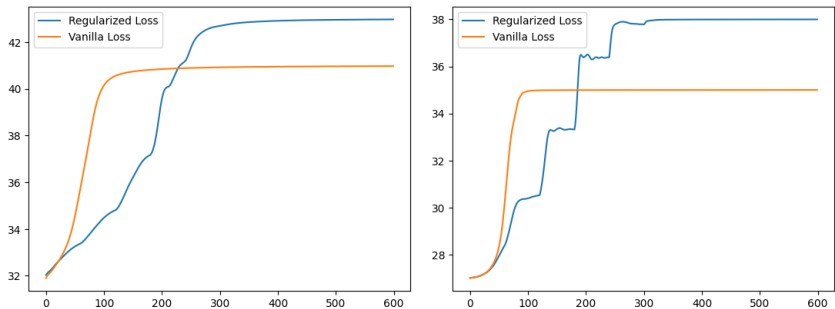

Figure 2: Plot of the Max-Cut value trajectory of the "vanilla" objective and entropy-regularized objective with the slow/fast mixture scheme. We remark that we plot the value of the cut of each iteration (and not the value of the regularized-loss). On the horizontal axis we plot the number of iterations and on the vertical axis we plot the achieved value of the cut. Both graphs used were random $G(n,p)$ graphs generated with $n = 15$ nodes and edge probabilidty $p = 0.5$. For the left plot we used a linear network (the same exponential family as the one used in our theoretical results). For the right plot we used a simple 3-Layer ReLU network to generate the scores. We observe that the "vanilla" loss gets stuck on sub-optimal solutions.

**Large Graph Instances.** A natural question is whether this improvement is also apparent in larger graphs. We focus on the case of random $d$-regular graphs with $n$ nodes. It is well-known that for this family of graphs, with high probability as $n \to \infty$, the size of the maximum cut satisfies $\mathrm{MaxCut}(G(n,d)) = n(d/4 + P_\star\sqrt{d/4} + o_d(\sqrt{d})) + o(n)$, where $P_\star \approx 0.7632$ is a universal constant [DMS17]. We aim to find a good approximation for the normalized cut-value, defined as $(\text{cut\_value}/n - d/4)/\sqrt{d/4}$, which (roughly speaking) takes values in $[0, P_\star]$. We obtain approximate random samples from the density $e^f$ using the Metropolis-Adjusted Langevin Algorithm (MALA). In particular, an approximate sample from this density is obtained by the Euler–Maruyama method for simulating the Langevin diffusion: $x_{k+1} = x_k + \tau\nabla f(x_k) + \sqrt{2\tau}\xi_k$, where $\xi_k$ is an independent Gaussian vector $\mathcal{N}(0, I)$. MALA incorporates an additional step based on the Metropolis-Hastings algorithm (see [Bes94, SK21]). In our case, the score function $f$ is a simple 3-layer ReLU network. In our experiments for 3 larger random regular graphs (600 nodes) using the fast/slow mixing technique along entropy regularization we see that our method leads to improvements over the vanilla objective. Plots of the trajectories of the vanilla and our method can be found in Figure 3. In the horizontal axis we plot the iterations and in the vertical axis we plot the normalized cut score of each method (higher is better) – we stop the plot of the vanilla trajectory after 200 iterations because we observed that its output has fully converged and is stuck.

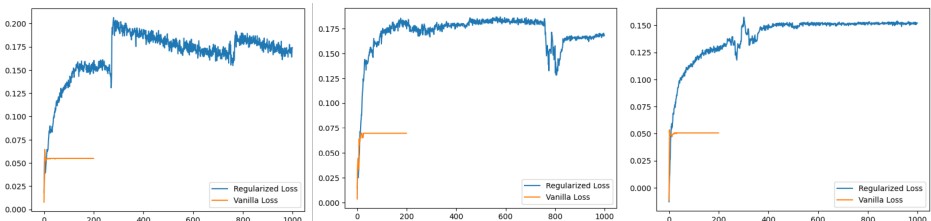

Figure 3: Plot of the Max-Cut value trajectory of the "vanilla" objective and entropy-regularized objective with the slow/fast mixture scheme on large instances. On the horizontal axis we plot the number of iterations and on the vertical axis we plot the achieved value of the normalized cut-value. The 3 graphs used were random $G(n,p)$ graphs generated with $n = 600$ nodes and edge probability $p = 0.5$. We stop the plot of the vanilla trajectory after 200 iterations because we observed that its output has fully converged and is stuck.

## Acknowledgements

This work has been partially supported by project MIS 5154714 of the National Recovery and Resilience Plan Greece 2.0 funded by the European Union under the NextGenerationEU Program.

Constantine Caramanis was partially supported by the NSF IFML Institute (NSF 2019844), and the NSF AI-EDGE Institute (NSF 2112471).

Dimitris Fotakis has been partially supported by the Hellenic Foundation for Research and Innovation (H.F.R.I.) under the "First Call for H.F.R.I. Research Projects to support Faculty members and Researchers and the procurement of high-cost research equipment grant", project BALSAM, HFRI-FM17-1424.

Christos Tzamos was partially supported by the NSF IFML Institute (NSF 2144298).

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

## A  Preliminaries and Notation

This lemma is a useful tool for quasar convex functions.

**Lemma 1** ([HMR16]). *Suppose that the functions $f_1, \ldots, f_n$ are individually $\gamma$-quasar convex in $X$ with respect to a common global minimum $\overline{x}$. Then for non-negative weights $a_1, \ldots, a_n$, the linear combination $f = \sum_{i \in [n]} a_i f_i$ is also $\gamma$-quasar convex with respect to $\overline{x}$ in $X$.*

In the proofs, we use the following notation: for a matrix $W$ and vectors $x, z$, we let

$$\phi(x; z; W) = \frac{\exp(z^\top W z)}{\sum_{y \in X} \exp(z^\top W y)}, \tag{1}$$

be a probability mass function over $X$ and we overload the notation as

$$\phi(x; w) = \frac{\exp(w \cdot x)}{\sum_{y \in X} \exp(w \cdot y)}. \tag{2}$$

## B  The Proof of Remark 1

*Proof.* Let $x_1, \ldots, x_n$ be the variables of the Max-Cut problem of interest and $S = \{-1, 1\}^n$ be the solution space. Consider $\mathcal{P}$ to be the collection of product distributions over $S$, i.e., for any $p \in \mathcal{P}$, it holds that, for any $s \in S$, $\mathbf{Pr}_{x \sim p}[x = s] = \prod_{i \in [n]} p_i^{\frac{1+s_i}{2}} (1 - p_i)^{\frac{1-s_i}{2}}$. Let us consider the cube $[\epsilon, 1 - \epsilon]^n$. This family is complete since the $O(\epsilon)$-sub-optimal solution of $I$ belongs to $\mathcal{P}$ and is compressed since the description size is $\text{poly}(n, \log(1/\epsilon))$. We show that in this setting there exist bad stationary points. Let $L_G$ be the Laplacian matrix of the input graph. For some product distribution $p \in \mathcal{P}$, it holds that

$$\mathcal{L}(p) = -\underset{x \sim p(\cdot)}{\mathbf{E}}[x^\top L_G x] = -(2p - 1)^\top \overline{L}_G (2p - 1), \quad \nabla_p \mathcal{L}(p) = -4\overline{L}_G (2p - 1),$$

where $\overline{L}_G$ is zero in the diagonal and equal to the Laplacian otherwise. Let us consider a vertex of the cube $p \in [\epsilon, 1 - \epsilon]^n$ which is highly and strictly sub-optimal, i.e., any single change of a node would strictly improve the number of edges in the cut and the score attained in $p$ is very large compared to $\min_{x \in S} -x^\top L_G x$. For any $i \in [n]$, we show that

$$(\nabla \mathcal{L}(p) \cdot e_i)((2p - 1) \cdot e_i) < 0.$$

This means that if $p_i$ is large (i.e., $1 - \epsilon$), then the $i$-th coordinate of the gradient of $\mathcal{L}(p)$ should be negative since this would imply that the negative gradient would preserve $p_i$ to the right boundary. Similarly for the case where $p_i$ is small. This means that this point is a stationary point and is highly sub-optimal by assumption.

Let $P$ (resp. $N$) be the set of indices in $[n]$ where $p$ takes the value $1 - \epsilon$ (resp. $\epsilon$). For any $i \in [n]$, let $\mathcal{N}(i)$ be its neighborhood in $G$. Let us consider $i \in P$. We have that $(2p - 1) \cdot e_i > 0$ and so it suffices to show that

$$(\overline{L}_G (2p - 1)) \cdot e_i > 0,$$

which corresponds to showing that

$$\sum_{j \in \mathcal{N}(i) \cap P} L_G(i, j)(1 - 2\epsilon) + \sum_{j \in \mathcal{N}(i) \cap N} L_G(i, j)(2\epsilon - 1) > 0,$$

and so we would like to have

$$\sum_{j \in P} L_G(i, j) - \sum_{j \in N} L_G(i, j) > 0.$$

Note that this is true for any $i \in [n]$ since the current solution is a strict local optimum. The same holds if $i \in N$. $\qquad\square$

## C Completeness

**Proposition 2** (Completeness). *Consider $\epsilon > 0$ and a prior $\mathcal{R}$ over $\mathcal{I}$. Assume that Assumption 1 holds. There exist $\beta^\star, \rho^\star \in (0,1)$ and $\mathcal{W}$ such that the family of solution generators $\mathcal{P}$ of Equation* (5) *is complete.*

*Proof.* Assume that $\mathcal{O}(s, I) = \psi_\mathcal{I}(I)^\top M \psi_S(s)$ and let $z = \psi_\mathcal{I}(I)$ and $x = \psi_S(s)$. Moreover, let $\alpha, C, D_S, D_\mathcal{I}$ be the parameters promised by Assumption 1. Let us consider the family $\mathcal{P} = \{p(W) : W \in \mathcal{W}\}$ with

$$p(x; z; W) = (1 - \beta^\star) \frac{e^{z^\top W x}}{\sum_{y \in X} e^{z^\top W y}} + \beta^\star \frac{e^{z^\top \rho^\star W x}}{\sum_{y \in X} e^{z^\top \rho^\star W y}} \,,$$

where the mixing weight $\beta^\star \in (0,1)$ and the inverse temperate $\rho^\star$ are to be decided. Recall that

$$\mathcal{L}(W) = \operatorname*{\mathbf{E}}_{z \sim \mathcal{R}} \operatorname*{\mathbf{E}}_{x \sim p(\cdot; z; W)} [L(x; z)] = \operatorname*{\mathbf{E}}_{z \sim \mathcal{R}} \operatorname*{\mathbf{E}}_{x \sim p(\cdot; z; W)} [z^\top M x] \,.$$

Let us pick the parameter matrix $W = -M/\lambda$. Let us now fix a $z \in \psi_\mathcal{I}(\mathcal{I})$. For the given matrix $M$, we can consider the finite set of values $V$ obtained by the quadratic forms $\{z^\top M x\}_{x \in \psi_S(S)}$. We further cluster these values so that they have distance at least $\epsilon$ between each other. We consider the level sets $C_i$ where $C_1$ is the subset of $S$ with minimum value $v_1(= v_1(z)) \in V$, $C_2$ is the subset with the second smallest $v_2(= v_2(z)) \in V$, etc. For fixed $z \in \psi_\mathcal{I}(\mathcal{I})$, we have that

$$\operatorname*{\mathbf{E}}_{x \sim p(\cdot; z; -M/\lambda)} [z^\top M x] = (1 - \beta^\star) \operatorname*{\mathbf{E}}_{x \sim \phi(\cdot; z; -M/\lambda)} [z^\top M x] + \beta^\star \operatorname*{\mathbf{E}}_{x \sim \phi(\cdot; z; -\rho^\star M/\lambda)} [z^\top M x] \,,$$

where $\phi$ comes from (1). We note that

$$\operatorname*{\mathbf{Pr}}_{x \sim \phi(\cdot; z; -M/\lambda)} [z^\top M x \in C_i] = \frac{|C_i| e^{-v_i/\lambda}}{\sum_j |C_j| e^{-v_j/\lambda}} \,.$$

We claim that, by letting $\lambda \to 0$, the above measure concentrates uniformly on $C_1$. The worst case scenario is when $|C_2| = |S| - |C_1|$ and $v_2 = v_1 + \epsilon$. Then we have that

$$\operatorname*{\mathbf{Pr}}_{x \sim \phi(\cdot; z; -M/\lambda)} [z^\top M x \in C_2] = \frac{|C_2|/|C_1| e^{(-v_2 + v_1)/\lambda}}{1 + |C_2|/|C_1| e^{(-v_2 + v_1)/\lambda}} \leq \delta \,,$$

when $1/\lambda > \log(|\psi_S(S)|/\delta)/\epsilon$, since in the worst case $|C_2|/|C_1| = \Omega(|\psi_S(S)|)$. Using this choice of $\lambda$ and taking expectation over $z$, we get that

$$\operatorname*{\mathbf{E}}_{z \sim \mathcal{R}} \operatorname*{\mathbf{E}}_{x \sim p(\cdot; z; -M/\lambda)} [z^\top M x] \leq (1 - \beta^\star) \operatorname*{\mathbf{E}}_{z \sim \mathcal{R}} \left[ (1 - \delta) \min_{x \in \psi_S(S)} L(x; z) + \delta v_2(z) \right] + \beta^\star \operatorname*{\mathbf{E}}_{x \sim \phi(\cdot; z; -\rho^\star M/\lambda)} [z^\top M x] \,.$$

First, we remark that by taking $\rho^\star = \operatorname{poly}(\alpha, 1/C, 1/D_S, 1/D_\mathcal{I})$, the last term in the right-hand side of the above expression can be replaced by the expected score of an almost-uniform solution (see Lemma 3 and Proposition 8), which is at most $\operatorname{poly}(D_S, D_\mathcal{I}, C) 2^{-|\psi_S(S)|}$ (and which is essentially negligible). Finally, one can pick $\beta^\star, \delta = \operatorname{poly}(\epsilon, 1/C, 1/D_S, 1/D_\mathcal{I})$ so that

$$\mathcal{L}(-M/\lambda) = \operatorname*{\mathbf{E}}_{z \sim \mathcal{R}} \operatorname*{\mathbf{E}}_{x \sim p(\cdot; z; -M/\lambda)} [z^\top M x] \leq \operatorname*{\mathbf{E}}_{z \sim \mathcal{R}} \left[ \min_{x \in \psi_S(S)} L(x; z) \right] + \epsilon \,.$$

This implies that $\mathcal{P}$ is complete by letting $\overline{W} = -M/\lambda \in \mathcal{W}$. This means that one can take $\mathcal{W}$ be a ball centered at 0 with radius (of $\epsilon$-sub-optimality) to be of order at least $B = \|M\|_\mathrm{F}/\lambda$. □

## D Compression

**Proposition 3** (Compression). *Consider $\epsilon > 0$ and a prior $\mathcal{R}$ over $\mathcal{I}$. Assume that Assumption 1 holds. There exist $\beta^\star, \rho^\star \in (0,1)$ and $\mathcal{W}$ such that the family of solution generators $\mathcal{P}$ of Equation* (5) *is compressed.*

*Proof.* We have that the bit complexity to represent the mixing weight $\beta^\star$ is $\operatorname{polylog}(D_S, D_\mathcal{I}, C, 1/\epsilon)$ and the description size of $\mathcal{W}$ is polynomial in $[\mathcal{I}]$ and in $\log(1/\epsilon)$. This follows from Assumption 1 since the feature dimensions $n_X$ and $n_Z$ are $\operatorname{poly}([\mathcal{I}])$ and $\mathcal{W}$ is a ball centered at 0 with radius $O(B)$, where $B = \|M\|_\mathrm{F}/\lambda \leq C/\lambda$, which are also $\operatorname{poly}([\mathcal{I}]/\epsilon)$. □

# E  Efficiently Optimizable

**Proposition 4** (Efficiently Optimizable). *Consider $\epsilon > 0$ and a prior $\mathcal{R}$ over $\mathcal{I}$. Assume that Assumption 1 holds. There exist $\beta^\star, \rho^\star \in (0, 1)$ and $\mathcal{W}$ such that family of solution generators $\mathcal{P}$ of Equation (5) is efficiently optimizable using Projected SGD, where the projection set is $\mathcal{W}$.*

The proof of this proposition is essentially decomposed into two parts: first, we show that the entropy-reularized loss of Equation (2) is quasar convex and then apply the projected SGD algorithm to $\mathcal{L}_\lambda$.

Recall that $H(W) = \mathbf{E}_{z \sim \mathcal{R}} \mathbf{E}_{x \sim \phi(\cdot; z; W)}[\log \phi(x; z; W)]$. Let $R$ be a weighted sum (to be in accordance with the mixture structure of $\mathcal{P}$) of negative entropy regularizers

$$R(W) = (1 - \beta^\star)H(W) + \frac{\beta^\star}{\rho^\star}H(\rho^\star W)\,, \tag{1}$$

where $\beta^\star, \rho^\star$ are the fixed parameters of $\mathcal{P}$ (recall Equation (5)). We define the regularized loss

$$\mathcal{L}_\lambda(W) = \mathcal{L}(W) + \lambda R(W)\,, \tag{2}$$

where

$$\mathcal{L}(W) = \mathbf{E}_{z \sim \mathcal{R}} \mathbf{E}_{x \sim p(\cdot; z; W)}[L(z; x)]\,, \quad p(W) \in \mathcal{P}\,.$$

## E.1  Quasar Convexity of the Regularized Loss

In this section, we show that $\mathcal{L}_\lambda$ of Equation (2) is quasar convex. We restate Proposition 1.

**Proposition 5** (Quasar Convexity). *Consider $\epsilon > 0$ and a prior $\mathcal{R}$ over $\mathcal{I}$. Assume that Assumption 1 holds. The function $\mathcal{L}_\lambda$ of Equation (2) with domain $\mathcal{W}$ is $\mathrm{poly}(C, D_S, D_\mathcal{I}, 1/\epsilon, 1/\alpha)$-quasar convex with respect to $-M/\lambda$ on the domain $\mathcal{W}$.*

*Proof.* We can write the loss $\mathcal{L}_\lambda$ as

$$\mathcal{L}_\lambda(W) = \mathbf{E}_{z \sim \mathcal{R}}[\mathcal{L}_{\lambda, z}(W)] = \mathbf{E}_{z \sim \mathcal{R}}[\mathcal{L}_z(W) + \lambda R_z(W)]\,,$$

where the mappings $\mathcal{L}_z$ and $R_z$ are instance-specific (i.e., we have fixed $z$). We can make use of Lemma 1, which states that linear combinations of quasar convex (with the same minimizer) remain quasar convex. Hence, since the functions $\mathcal{L}_{\lambda, z}$ have the same minimizer $-M/\lambda$, it suffices to show quasar convexity for a particular fixed instance mapping, i.e., it sufffices to show that the function

$$\mathcal{L}_{\lambda, z}(W) = \mathcal{L}_z(W) + \lambda R_z(W)$$

is quasar convex. Recall that $W$ is a matrix of dimension $n_Z \times n_X$. To deal with the function $\mathcal{L}_{\lambda, z}$, we consider the simpler function that maps vectors instead of matrices to real numbers. For some vector $c$, let $\mathcal{L}_\lambda^{\mathrm{vec}} : \mathbb{R}^{n_X} \to \mathbb{R}$ be

$$\mathcal{L}_\lambda^{\mathrm{vec}}(w) = \mathbf{E}_{x \sim p(\cdot; w)}[c \cdot x] + \lambda R^{\mathrm{vec}}(w)\,, \tag{3}$$

where for any vector $w \in \mathbb{R}^{n_X}$, we define the probability distribution $\phi(\cdot; w)$ over the solution space $X = \psi_S(S)$ with probability mass function

$$\phi(x; w) = \frac{e^{w \cdot x}}{\sum_{y \in X} e^{w \cdot y}}\,.$$

We then define

$$p(\cdot; w) = (1 - \beta^\star)\phi(\cdot; w) + \beta^\star\phi(\cdot; \rho^\star w)\,,$$

and $R^{\mathrm{vec}}(w) = (1 - \beta^\star)H(w) + \frac{\beta^\star}{\rho^\star}H(\rho^\star w)$ (this is a essentially a weighted sum of regularizers, needed to simplify the proof) with $H(w) = \mathbf{E}_{x \sim \phi(\cdot; w)} \log \phi(x, w)$. These quantities are essentially the fixed-instance analogues of Equations (5) and (2). The crucial observation is that by taking $c = z^\top M$ and applying the chain rule we have that

$$\nabla_W \mathcal{L}_{\lambda, z}(W) = z \cdot \left[\nabla_w \mathcal{L}_\lambda^{\mathrm{vec}}(z^\top W)\right]^\top\,. \tag{4}$$

This means that the gradient of the fixed-instance objective $\mathcal{L}_{\lambda,z}$ is a matrix of dimension $n_Z \times n_X$ that is equal to the outer product of the instance featurization $z$ and the gradient of the simpler function $\mathcal{L}_w^{\text{vec}}$ evaluated at $z^\top W$. Let us now return on showing that $\mathcal{L}_{\lambda,z}$ is quasar convex. To this end, we observe that

$$\nabla_W \mathcal{L}_{\lambda,z}(W) \cdot \left(W + \frac{M}{\lambda}\right) = \nabla_w \mathcal{L}_\lambda^{\text{vec}}(z^\top W) \cdot \left(z^\top W + z^\top \frac{M}{\lambda}\right).$$

This means that, since $z$ is fixed, it suffices to show that the function $\mathcal{L}_\lambda^{\text{vec}}$ is quasar convex. We provide the next key proposition that deals with issue. This result is one the main technical aspects of this work and its proof can be found in Appendix E.2.

In the following, intuitively $\mathcal{X}$ is the post-featurization instance space and $\mathcal{Z}$ is the parameter space.

**Proposition 6.** *Consider $\epsilon, \lambda > 0$. Let $\|c\|_2 \leq C_1$. Let $\mathcal{Z}$ be an open ball centered at 0 with diameter $2C_1/\lambda$. Let $\mathcal{X}$ be a space of diameter $D$ and let $\mathbf{Var}_{x\sim U(\mathcal{X})}[v \cdot x] \geq \alpha\|v\|_2^2$ for any $v \in \mathcal{Z}$. The function $\mathcal{L}_\lambda^{\text{vec}}(w) = \mathbf{E}_{x\sim p(\cdot;w)}[c \cdot x] + \lambda R^{\text{vec}}(w)$ is $\text{poly}(1/C_1, 1/D, \epsilon, \alpha)$-quasar convex with respect to $-c/\lambda$ on $\mathcal{Z}$.*

We can apply the above result with $c = z^\top M, w = z^\top W, D = D_S$ and $C_1 = D_\mathcal{I} C$. These give that the quasar convexity parameter $\gamma$ is of order $\gamma = \text{poly}(\epsilon, \alpha, 1/C, 1/D_S, 1/D_\mathcal{I})$. Since we have that $\mathcal{L}_\lambda^{\text{vec}}(z^\top W) = \mathcal{L}_{\lambda,z}(W)$, we get that

$$\nabla_W \mathcal{L}_{\lambda,z}(W) \cdot (W + M/\lambda) \geq \gamma(\mathcal{L}_{\lambda,z}(W) - \mathcal{L}_{\lambda,z}(-M/\lambda)).$$

This implies that $\mathcal{L}_{\lambda,z}$ is $\gamma$-quasar convex with respect to the minimizer $-M/\lambda$ and completes the proof using Lemma 1. □

## E.2 The Proof of Proposition 6

Let us consider $\mathcal{L}_\lambda^{\text{vec}}$ to be a real-valued differentiable function defined on $\mathcal{Z}$. Let $w, -c/\lambda \in \mathcal{Z}$ and let $L$ be the line segment between them with $L \in \mathcal{Z}$. The mean value theorem implies that there exists $w' \in L$ such that

$$\mathcal{L}_\lambda^{\text{vec}}(w) - \mathcal{L}_\lambda^{\text{vec}}(-c/\lambda) = \nabla_w \mathcal{L}_\lambda^{\text{vec}}(w') \cdot (w + c/\lambda) \leq \|\nabla \mathcal{L}_\lambda^{\text{vec}}(w')\|_2 \|w + c/\lambda\|_2.$$

Now we have that $\mathcal{L}_\lambda^{\text{vec}}$ has bounded gradient (see Lemma 2) and so we get that

$$\|\nabla_w \mathcal{L}_\lambda^{\text{vec}}(w')\|_2 \leq D^2 \|c + \lambda w'\|_2 = D^2 \lambda \|w' + c/\lambda\|_2 \leq D^2 \lambda \|w + c/\lambda\|_2,$$

since $w' \in L$. This implies that

$$\mathcal{L}_\lambda^{\text{vec}}(w) - \mathcal{L}_\lambda^{\text{vec}}(-c/\lambda) \leq D^2 \lambda \|w + c/\lambda\|_2^2 \leq \frac{1}{\gamma} \nabla \mathcal{L}_\lambda^{\text{vec}}(w) \cdot (w + c/\lambda),$$

where $1/\gamma = \frac{\text{poly}(C_1, D)}{\epsilon^3 \alpha^2}$. The last inequality is an application of the correlation lower bound (see Lemma 3).

In the above proof, we used two key lemmas: a bound for the norm of the gradient and a lower bound for the correlation. In the upcoming subsections, we prove these two results.

### E.2.1 Bounded Gradient Lemma and Proof

**Lemma 2** (Bounded Gradient Norm of $\mathcal{L}_\lambda^{\text{vec}}$). *Consider $\epsilon, \lambda > 0$. Let $\mathcal{Z}$ be the domain of $\mathcal{L}_\lambda^{\text{vec}}$ of (3). Let $\mathcal{X}$ be a space of diameter $D$. For any $w \in \mathcal{Z}$, it holds that*

$$\|\nabla_w \mathcal{L}_\lambda^{\text{vec}}(w)\|_2 \leq O(D^2)\|c + \lambda w\|_2.$$

*Proof.* We have that

$$\nabla_w \mathcal{L}_\lambda^{\text{vec}}(w) = (1 - \beta^\star)G_w + \beta^\star \rho^\star G_{\rho^\star w},$$

where

$$G_w = \mathop{\mathbf{E}}_{x\sim\phi(\cdot;w)}[((c + \lambda w) \cdot x)x] - \mathop{\mathbf{E}}_{x\sim\phi(\cdot;w)}[(c + \lambda w) \cdot x] \mathop{\mathbf{E}}_{x\sim\phi(\cdot;w)}[x],$$

and
$$G_{\rho^\star w} = \underset{x \sim \phi(\cdot; \rho^\star w)}{\mathbf{E}}[((c + \lambda w) \cdot x)x] - \underset{x \sim \phi(\cdot; \rho^\star w)}{\mathbf{E}}[(c + \lambda w) \cdot x] \underset{x \sim \phi(\cdot; \rho^\star w)}{\mathbf{E}}[x].$$

Note that since $x \in \mathcal{X}$, it holds that $\|x\|_2 \leq D$. Hence

$$\|G_w\|_2 = \sup_{v: \|v\|_2 = 1} |v \cdot G_w| \leq 2D^2 \|c + \lambda w\|_2.$$

Moreover, we have that

$$\|G_{\rho^\star w}\|_2 \leq 2D^2 \|c + \lambda w\|_2.$$

This means that

$$\|\nabla_w \mathcal{L}_\lambda^{\mathrm{vec}}(w)\|_2 \leq 2(1 - \beta^\star)\|c + \lambda w\|_2 D^2 + 2\beta^\star \rho^\star \|c + \lambda w\|_2 D^2 = O(D^2)\|c + \lambda w\|_2.$$

$\square$

### E.2.2 Correlation Lower Bound Lemma and Proof

The following lemma is the second ingredient in order to show Proposition 6.

**Lemma 3** (Correlation Lower Bound for $\mathcal{L}_\lambda^{\mathrm{vec}}$). *Let $\lambda > 0$. Let $\|c\|_2 \leq C_1$. Let $\mathcal{Z}$ be an open ball centered at 0 with diameter $B = 2C_1/\lambda$. Let $\mathcal{X}$ be a space of diameter $D$. Assume that $w \in \mathcal{Z}$ and $\mathbf{Var}_{x \sim U(\mathcal{X})}[(c + \lambda w) \cdot x] \geq \alpha \|c + \lambda w\|_2^2$ for some $\alpha > 0$. Then, for any $\beta^\star \in (0, 1)$, there exists $\rho^\star > 0$ such that it holds that*

$$\nabla_w \mathcal{L}_\lambda^{\mathrm{vec}}(w) \cdot (c + \lambda w) = \Omega\left(\beta^\star \alpha^2/(BD^3)\|c + \lambda w\|_2^2\right),$$

*where $\mathcal{L}_\lambda^{\mathrm{vec}}$ is the regularized loss of Proposition 6, $\rho^\star$ is the scale in the second component of the mixture of (5) and $\beta^\star \in (0, 1)$ is the mixture weight.*

First, in Lemma 4 and Lemma 5, we give a formula for the desired correlation $\nabla_w \mathcal{L}_\lambda^{\mathrm{vec}}(w) \cdot (c + \lambda w)$ and, then we can provide a proof for Lemma 3 by lower bounding this formula.

**Lemma 4** (Correlation with Regularization). *Consider the function $g(w) = \mathbf{E}_{x \sim \phi(\cdot; w)}[c \cdot x] + \lambda H(w)$, where $H$ is the negative entropy regularizer. Then it holds that*

$$\nabla_w g(w) \cdot (c + \lambda w) = \mathbf{Var}_{x \sim \phi(\cdot; w)}[(c + \lambda w) \cdot x].$$

*Proof.* Let us consider the following objective function:

$$g(w) = \underset{x \sim \phi(\cdot; w)}{\mathbf{E}}[c \cdot x] + \lambda H(w), \quad \phi(x; w) = \frac{\exp(w \cdot x)}{\sum_{y \in \mathcal{X}} \exp(w \cdot y)},$$

where $H$ is the negative entropy regularizer, i.e.,

$$H(w) = \underset{x \sim \phi(\cdot; w)}{\mathbf{E}}[\log \phi(x; w)] = \underset{x \sim \phi(\cdot; w)}{\mathbf{E}}[w \cdot x] - \log\left(\sum_{y \in \mathcal{X}} e^{w \cdot y}\right).$$

The gradient of $g$ with respect to $w \in \mathcal{W}$ is equal to

$$\nabla_w g(w) = \underset{x \sim \phi(\cdot; w)}{\mathbf{E}}[(c \cdot x)\nabla_w \log \phi(x; w)] + \lambda \nabla_w H(w).$$

It holds that

$$\nabla_w \log \phi(x; w) = \nabla_w \left(w \cdot x - \log \sum_{y \in \mathcal{X}} e^{w \cdot y}\right) = x - \underset{x \sim \phi(\cdot; w)}{\mathbf{E}}[x],$$

and

$$\nabla H(w) = \underset{x \sim \phi(\cdot; w)}{\mathbf{E}}[x] + \underset{x \sim \phi(\cdot; w)}{\mathbf{E}}[(w \cdot x)\nabla_w \log \phi(x; w)] - \underset{x \sim \phi(\cdot; w)}{\mathbf{E}}[x] = \underset{x \sim \phi(\cdot; w)}{\mathbf{E}}[(w \cdot x)\nabla_w \log \phi(x; w)].$$

So, we get that

$$\nabla_w g(w) = \underset{x \sim \phi(\cdot; w)}{\mathbf{E}}[((c + \lambda w) \cdot x)x] - \underset{x \sim \phi(\cdot; w)}{\mathbf{E}}[(c + \lambda w) \cdot x] \underset{x \sim \phi(\cdot; w)}{\mathbf{E}}[x].$$

Note that

$$\nabla_w g(w) \cdot (c + \lambda w) = \mathbf{Var}_{x \sim \phi(\cdot; w)}[(c + \lambda w) \cdot x].$$

$\square$

**Lemma 5** (Gradient with Regularization and Mixing). *For any $\epsilon > 0$, for the family of solution generators $\mathcal{P} = \{p(\cdot; w) = (1 - \beta^\star)\phi(\cdot; w) + \beta^\star \phi(\cdot; \rho^\star w) : w \in \mathcal{W}\}$ and the objective $\mathcal{L}_\lambda^{\mathrm{vec}}$ of Equation (3), it holds that*

$$\nabla_w \mathcal{L}_\lambda^{\mathrm{vec}}(w) \cdot (c + \lambda w) = (1 - \beta^\star) \mathbf{Var}_{x \sim \phi(\cdot; w)}[(c + \lambda w) \cdot x] + \beta^\star \rho^\star \mathbf{Var}_{x \sim \phi(\cdot; \rho^\star w)}[(c + \lambda w) \cdot x],$$

*for any $\lambda > 0$.*

*Proof.* Let us first consider the scaled parameter $\rho w \in \mathcal{W}$ for some $\rho > 0$. Then it holds that

$$\nabla_w \mathop{\mathbf{E}}_{x \sim \phi(\cdot; \rho w)}[c \cdot x] = \mathop{\mathbf{E}}_{x \sim \phi(\cdot; \rho w)}\left[ (c \cdot x)\left( \rho x - \mathop{\mathbf{E}}_{x \sim \phi(\cdot; \rho w)}[\rho x] \right) \right] = \rho \mathop{\mathbf{E}}_{x \sim \phi(\cdot; \rho w)}\left[ (c \cdot x)\left( x - \mathop{\mathbf{E}}_{x \sim \phi(\cdot; \rho w)}[x] \right) \right].$$

Moreover, the negative entropy regularizer at $\rho w$ is

$$H(\rho w) = \mathop{\mathbf{E}}_{x \sim \phi(\cdot; \rho w)}[(\rho w) \cdot x] - \log \sum_{y \in \mathcal{X}} e^{(\rho w) \cdot y}.$$

It holds that

$$\nabla_w H(\rho w) = \mathop{\mathbf{E}}_{x \sim \phi(\cdot; \rho w)}[(\rho w \cdot x)\nabla_w \log \phi(x; \rho w)] = \rho^2 \left( \mathop{\mathbf{E}}_{x \sim \phi(\cdot; \rho w)}[(w \cdot x)x] - \mathop{\mathbf{E}}_{x \sim \phi(\cdot; \rho w)}[w \cdot x] \mathop{\mathbf{E}}_{x \sim \phi(\cdot; \rho w)}[x] \right).$$

We consider the objective function $\mathcal{L}_\lambda^{\mathrm{vec}}$ to be defined as follows: first, we take

$$p(\cdot; w) = (1 - \beta^\star)\phi(\cdot; w) + \beta^\star \phi(\cdot; \rho^\star w),$$

i.e., $p(\cdot; w)$ is the mixture of the probability measures $\phi(\cdot; w)$ and $\phi(\cdot; \rho^\star w)$ with weights $1 - \beta^\star$ and $\beta^\star$ respectively for some scale $\rho^\star > 0$. Moreover, we take $R^{\mathrm{vec}}(w) = (1 - \beta^\star)H(w) + \frac{\beta^\star}{\rho^\star}H(\rho^\star w)$. Then we define our regularized loss $\mathcal{L}_\lambda^{\mathrm{vec}}$ to be

$$\mathcal{L}_\lambda^{\mathrm{vec}}(w) = \mathop{\mathbf{E}}_{x \sim p(\cdot; w)}[c \cdot x] + \lambda R^{\mathrm{vec}}(w).$$

Using Lemma 4 and the above calculations, we have that

$$\nabla_w \mathcal{L}_\lambda^{\mathrm{vec}}(w) = (1 - \beta^\star)\nabla_w \mathop{\mathbf{E}}_{x \sim \phi(\cdot; w)}[c \cdot x] + \lambda(1 - \beta^\star)\nabla_w H(w) + \beta^\star \nabla_w \mathop{\mathbf{E}}_{x \sim \phi(\cdot; \rho^\star x)}[c \cdot x] + \lambda \frac{\beta^\star}{\rho^\star}\nabla_w H(\rho^\star w)$$

$$= (1 - \beta^\star)\left( \mathop{\mathbf{E}}_{x \sim \phi(\cdot; w)}[((c + \lambda w) \cdot x)x] - \mathop{\mathbf{E}}_{x \sim \phi(\cdot; w)}[(c + \lambda w) \cdot x] \mathop{\mathbf{E}}_{x \sim \phi(\cdot; w)}[x] \right) +$$

$$+ \beta^\star \rho^\star \left( \mathop{\mathbf{E}}_{x \sim \phi(\cdot; \rho^\star w)}[((c + \lambda w) \cdot x)x] - \mathop{\mathbf{E}}_{x \sim \phi(\cdot; \rho^\star w)}[(c + \lambda w) \cdot x] \mathop{\mathbf{E}}_{x \sim \phi(\cdot; \rho^\star w)}[x] \right).$$

The above calculations yield

$$\nabla_w \mathcal{L}_\lambda^{\mathrm{vec}}(w) \cdot (c + \lambda w) = (1 - \beta^\star) \mathbf{Var}_{x \sim \phi(\cdot; w)}[(c + \lambda w) \cdot x] + \beta^\star \rho^\star \mathbf{Var}_{x \sim \phi(\cdot; \rho^\star w)}[(c + \lambda w) \cdot x],$$

and this concludes the proof. $\qquad\square$

The above correlation being positive intuitively means that performing gradient descent to $\mathcal{L}_\lambda^{\mathrm{vec}}$ gives that the parameter $w$ converges to $-c/\lambda$, the point that achieves completeness for that objective.

However, to obtain fast convergence, we need to show that the above correlation is non-trivial. This means that our goal in order to prove Lemma 3 is to provide a lower bound for the above quantity, i.e., it suffices to give a non-trivial lower bound for the variance of the random variable $(c + \lambda w) \cdot x$ with respect to the probability measure $\phi(\cdot; \rho^\star w)$. It is important to note that in the above statement we did not fix the value of $\rho^\star$. We can now make use of Proposition 8. Intuitively, by taking the scale parameter appearing in the mixture $\rho^\star$ to be sufficiently small, we can manage to provide a lower bound for the variance of $(c + \lambda w) \cdot x$ with respect to the almost uniform measure, i.e, the second summand of the above right-hand side expression has significant contribution. We remark that $\rho$ corresponds to the inverse temperature parameter. Hence, our previous analysis essentially implies that policy gradient on combinatorial optimization potentially works if the variance $\mathbf{Var}_{x \sim \phi(\cdot; \rho w)}[(c + \lambda w) \cdot x]$ is non-vanishing at high temperatures $1/\rho$.

*The proof of Lemma 3.* Recall that

$$\phi(x; w) = \frac{e^{w \cdot x}}{\sum_{y \in \mathcal{X}} e^{w \cdot y}}.$$

Let also $D$ be the diameter of $\mathcal{X}$ and $B$ the diameter of $\mathcal{Z}$. Recall from Lemma 5 that we have that

$$\nabla_w \mathcal{L}_\lambda^{\text{vec}}(w) \cdot (c + \lambda w) = (1 - \beta^\star) \mathbf{Var}_{x \sim p(\cdot; w)}[(c + \lambda w) \cdot x] + \beta^\star \rho^\star \mathbf{Var}_{x \sim p(\cdot; \rho^\star w)}[(c + \lambda w) \cdot x],$$

where the scale parameter $\rho^\star > 0$ is to be decided. This means that

$$\nabla_w \mathcal{L}_\lambda^{\text{vec}}(w) \cdot (c + \lambda w) \geq \beta^\star \rho^\star \mathbf{Var}_{x \sim \phi(\cdot; \rho^\star w)}[(c + \lambda w) \cdot x].$$

Our goal is now to apply Proposition 8 in order to lower bound the above variance. Applying Proposition 8 for $\mu \leftarrow \phi(\cdot; \rho^\star w), c \leftarrow c + \lambda w \in \mathcal{Z}$ and, so for some absolute constant $C_0$, we can pick

$$\rho^\star = C_0 \frac{\alpha}{BD^3}.$$

Thus, we have that

$$\mathbf{Var}_{x \sim \phi(\cdot; \rho^\star w)}[(c + \lambda w) \cdot x] \geq \Omega(\alpha \|c + \lambda w\|_2^2).$$

This implies the desired result since

$$\nabla_w \mathcal{L}_\lambda^{\text{vec}}(w) \cdot (c + \lambda w) \geq C_0 \beta^\star \alpha^2 / (BD^3) \|c + \lambda w\|_2^2.$$

$\square$

### E.3 Convergence for Quasar Convex Functions

The fact that $\mathcal{L}_\lambda$ is quasar convex with respect to $-M/\lambda$ implies that projected SGD converges to that point in a small number if steps and hence the family $\mathcal{P}$ is efficiently optimizable. The analysis is standard (see e.g., [HMR16]). For completeness a proof can be found in Appendix E.3.

**Proposition 7** (Convergence). *Consider $\epsilon > 0$ and a prior $\mathcal{R}$ over $\mathcal{I}$. Assume that Assumption 1 holds with parameters $C, D_S, D_{\mathcal{I}}, \alpha$. Let $W_1, \ldots, W_T$ be the updates of the SGD algorithm with projection set $\mathcal{W}$ performed on $\mathcal{L}_\lambda$ of Equation (2) with appropriate step size and parameter $\lambda$. Then, for the non-regularized objective $\mathcal{L}$, it holds that*

$$\mathbf{E}_{t \sim U([T])}[\mathcal{L}(W_t)] \leq \mathcal{L}(-M/\lambda) + \epsilon,$$

*when $T \geq \text{poly}(1/\epsilon, 1/\alpha, C, D_S, D_{\mathcal{I}}, \|W_0 + M/\lambda\|_{\text{F}})$.*

Our next goal is to use Proposition 1 and show that standard projected SGD on the objective $\mathcal{L}_\lambda$ converges in a polynomial number of steps. The intuition behind this result is that since the correlation between $\nabla \mathcal{L}_\lambda(W)$ and the direction $M + \lambda W$ is positive and non-trivial, the gradient field drives the optimization method towards the point $-M/\lambda$.

*Proof.* Consider the sequence of matrices $W_1, \ldots, W_t, \ldots, W_T$ generated by applying PSGD on $\mathcal{L}_\lambda$ with step size $\eta$ (to be decided) and initial parameter vector $W_0 \in \mathcal{W}$. We have that $\mathcal{L}_\lambda$ is $\gamma$-quasar convex and is also $O(\Gamma)$-weakly smooth[2] since we now show that it is $\Gamma$-smooth.

**Lemma 6.** $\mathcal{L}_\lambda$ *is* $\text{poly}(D_S, D_{\mathcal{I}}, C)$*-smooth.*

*Proof.* We have that

$$\|\nabla \mathcal{L}_\lambda(W)\|_{\text{F}}^2 = \|\mathbf{E}_{z \sim \mathcal{R}}[\nabla \mathcal{L}_{\lambda, z}(W)]\|_{\text{F}}^2 \leq \mathbf{E}_{z \sim \mathcal{R}} \|z\|_2^2 \|\nabla_w \mathcal{L}_\lambda^{\text{vec}}(z^\top W)\|_{\text{F}}^2.$$

It suffices to show that $\mathcal{L}_\lambda^{\text{vec}}$ is smooth. Recall that

$$\nabla_w \mathcal{L}_\lambda^{\text{vec}}(w) = (1 - \beta^\star) \left( \mathbf{E}_{x \sim \phi(\cdot; w)}[((c + \lambda w) \cdot x)x] - \mathbf{E}_{x \sim \phi(\cdot; w)}[(c + \lambda w) \cdot x] \mathbf{E}_{x \sim \phi(\cdot; w)}[x] \right) +$$

$$+ \beta^\star \rho^\star \left( \mathbf{E}_{x \sim \phi(\cdot; \rho^\star w)}[((c + \lambda w) \cdot x)x] - \mathbf{E}_{x \sim \phi(\cdot; \rho^\star w)}[(c + \lambda w) \cdot x] \mathbf{E}_{x \sim \phi(\cdot; \rho^\star w)}[x] \right).$$

---

[2]As mentioned in [HMR16], a function $f$ is $\Gamma$-weakly smooth if for any point $\theta$, $\|\nabla f(\theta)\|^2 \leq \Gamma(f(\theta) - f(\theta^\star))$. Moreover, a function $f$ that is $\Gamma$-smooth (in the sense $\|\nabla^2 f\| \leq \Gamma$), is also $O(\Gamma)$-weakly smooth.

This means that

$$\|\nabla_w^2 \mathcal{L}_\lambda^{\text{vec}}(w)\|_{\mathrm{F}}^2 \leq (1 - \beta^\star)(A_1 + A_2) + \beta^\star \rho^\star (A_3 + A_4),$$

where

$$A_1 = \left\|\nabla_w \mathop{\mathbf{E}}_{x \sim \phi(\cdot;w)}[((c + \lambda w) \cdot x)x]\right\|_{\mathrm{F}}^2, \quad A_2 = \left\|\nabla_w \mathop{\mathbf{E}}_{x \sim \phi(\cdot;w)}[(c + \lambda w) \cdot x] \mathop{\mathbf{E}}_{x \sim \phi(\cdot;w)}[x]\right\|_{\mathrm{F}}^2,$$

$$A_3 = \left\|\nabla_w \mathop{\mathbf{E}}_{x \sim \phi(\cdot;\rho^\star w)}[((c + \lambda w) \cdot x)x]\right\|_{\mathrm{F}}^2, \quad A_4 = \left\|\nabla_w \mathop{\mathbf{E}}_{x \sim \phi(\cdot;\rho^\star w)}[(c + \lambda w) \cdot x] \mathop{\mathbf{E}}_{x \sim \phi(\cdot;\rho^\star w)}[x]\right\|_{\mathrm{F}}^2.$$

Standard computation of these values yields that, since $D_S$ and $D_{\mathcal{I}}$ are bounds to $x$ and $z$ respectively, we have that $\mathcal{L}_\lambda$ is smooth with parameter $\text{poly}(D_S, D_{\mathcal{I}}, C)$. □

Let $V$ be the variance of the unbiased estimator used for $\nabla_W \mathcal{L}_\lambda(W)$. We can apply the next result of [HMR16].

**Lemma 7** ([HMR16]). *Suppose the objective function $f$ is $\gamma$-weakly quasi convex and $\Gamma$-weakly smooth, and let $r(\cdot)$ be an unbiased estimator for $\nabla f(\theta)$ with variance $V$. Moreover, suppose the global minimum $\overline{\theta}$ belongs to $\mathcal{W}$, and the initial point $\theta_0$ satisfies $\|\theta_0 - \overline{\theta}\|_2 \leq R$. Then projected stochastic gradient descent with a proper learning rate returns $\theta_T$ in $T$ iterations with expected error*

$$\mathop{\mathbf{E}}_{t \sim U([T])} f(\theta_t) - f(\overline{\theta}) \leq \max\left\{\frac{\Gamma R^2}{\gamma^2 T}, \frac{R\sqrt{V}}{\gamma\sqrt{T}}\right\}.$$

We apply the above result to $\mathcal{L}_\lambda$ in order to find matrices $W_1, ..., W_T$ that achieve good loss on average compared to $-M/\lambda$. Moreover, using a batch SGD update, we can take $V$ to be also polynomial in the crucial parameters of the problem. We note that one can adapt the above convergence proof and show that the actual loss $\mathcal{L}$ (and not the loss $\mathcal{L}_\lambda$) are close after sufficiently many iterations (as indicated by the above lemma). We know that the Frobenius norm of the gradient of $\mathcal{L}(W)$ is at most of order $O(D_{\mathcal{I}}^2 C D_S^2)$. We can apply the mean value theorem in high dimensions (by taking $\mathcal{W}$ to be an open ball of radius $O(B)$) and this yields that the difference between the values of $\mathcal{L}(W_T)$ and $\mathcal{L}(-M/\lambda)$ is at most $D_{\mathcal{I}}^2 C D_S^2 \|W_t + M/\lambda\|_{\mathrm{F}}^2$. However, the right-hand side is upper bounded by the correlation between $\nabla \mathcal{L}_\lambda(W_t)$ and $W_t + M/\lambda$. Hence, we can still use this correlation as a potential in order to minimize $\mathcal{L}$. This implies that the desired convergence guarantee holds as long as $T \geq \text{poly}(1/\epsilon, 1/\alpha, C, D_S, D_{\mathcal{I}}, \|W_0 + M/\lambda\|_{\mathrm{F}})$. □

## F Deferred Proofs: Variance under Almost Uniform Distributions

This section is a technical section that states some properties of exponential families. We use some standard notation, such as $w$ and $x$, for the statements and the proofs but we underline that these symbols do not correspond to the notation in the main body of the paper.

We consider the parameter space $\Theta$ and for any parameter $w \in \Theta$, we define the probability distribution $\phi(\cdot; w)$ over a space $\mathcal{X}$ with density

$$\phi(x; w) = \frac{e^{w \cdot x}}{\sum_{y \in \mathcal{X}} e^{w \cdot y}}.$$

In this section, our goal is to relate the variance of $c \cdot x$ under the measure $\phi(\cdot; 0)$ (uniform case) and $\phi(\cdot; \rho^\star w)$ for some $w \in \mathcal{W}$ and some sufficiently small $\rho^\star$ (almost uniform case). The main result of this section follows.

**Proposition 8** (Variance Lower Bound Under Almost Uniform Distributions). *Assume that the variance of $c \cdot x$ under the uniform distribution over $\mathcal{X}$, whose diameter is $D$, is lower bounded by $\alpha\|c\|_2^2$. Moreover assume that $w \in \Theta$ with $\|w\|_2 \leq B$. Then, setting $\rho^\star = O(\alpha/(BD^3))$, it holds that $\mathbf{Var}_{x \sim \phi(\cdot;\rho^\star w)}[c \cdot x] = \Omega(\alpha\|c\|_2^2)$.*

We first provide a general abstract lemma that relates the variance of the uniform distribution $U$ over $\mathcal{X}$ to the variance of an almost uniform probability measure $\mu$. For simplicity, we denote the uniform distribution over $\mathcal{X}$ with $U = U(\mathcal{X})$.

**Lemma 8.** *Let $w \in \Theta$ and $x \in \mathcal{X}$ with $\|x\|_2 \leq D$. Consider the uniform probability measure $U$ over $\mathcal{X}$ and let $\mu$ over $\mathcal{X}$ be such that there exist $\epsilon_1, \epsilon_2 > 0$ with:*

- $\|\mathbf{E}_{x \sim U}[x] - \mathbf{E}_{x \sim \mu}[x]\|_2 \leq \epsilon_1$, *and,*

- $w^\top \mathbf{E}_{x \sim \mu}[xx^\top]w \geq w^\top \mathbf{E}_{x \sim U}[xx^\top]w - \epsilon_2 \|w\|_2^2.$

*Then it holds that* $\mathbf{Var}_{x \sim \mu}[w \cdot x] \geq \mathbf{Var}_{x \sim U}[w \cdot x] - 3\max\{\epsilon_1^2, \epsilon_1 D, \epsilon_2\}\|w\|_2^2.$

*Proof.* We have that

$$\mathbf{Var}_{x \sim \mu}[w \cdot x] = \mathbf{E}_{x \sim \mu}\left[(w \cdot x)^2\right] - \left(\mathbf{E}_{x \sim \mu}[w \cdot x]\right)^2.$$

We first deal with upper-bounding the square of the first moment. Note that

$$w \cdot \left(\mathbf{E}_{x \sim \mu}[x] - \mathbf{E}_{x \sim U}[x]\right) \leq \|w\|_2 \left\|\mathbf{E}_{x \sim \mu}[x] - \mathbf{E}_{x \sim U}[x]\right\|_2 \leq \epsilon_1 \|w\|_2.$$

Let us take $\epsilon > 0$ (with $\epsilon < \epsilon_1$) for simplicity to be such that $\left(\mathbf{E}_{x \sim \mu}[w \cdot x]\right)^2 = \left(\mathbf{E}_{x \sim U}[w \cdot x] + \epsilon\|w\|_2\right)^2$. This means that

$$\left(\mathbf{E}_{x \sim \mu}[w \cdot x]\right)^2 \leq \left(\mathbf{E}_{x \sim U}[w \cdot x]\right)^2 + 2\epsilon\|w\|_2 \left|\mathbf{E}_{x \sim U}[w \cdot x]\right| + \epsilon^2\|w\|_2^2$$

$$\leq \left(\mathbf{E}_{x \sim U}[w \cdot x]\right)^2 + 2\epsilon D\|w\|_2^2 + \epsilon^2\|w\|_2^2.$$

Next we lower-bound the second moment. It holds that

$$\mathbf{E}_{x \sim \mu}\left[(w \cdot x)^2\right] = w^\top \mathbf{E}_{x \sim \mu}\left[xx^\top\right]w \geq \mathbf{E}_{x \sim U}[(w \cdot x)^2] - \epsilon_2\|w\|_2^2,$$

for some $\epsilon_2 > 0$. This means that

$$\mathbf{Var}_{x \sim \mu}(w \cdot x) \geq \mathbf{E}_{x \sim U}\left[(w \cdot x)^2\right] - \epsilon_2\|w\|_2^2 - \left(\mathbf{E}_{x \sim U}[w \cdot x]\right)^2 - 2\epsilon D\|w\|_2^2 - \epsilon^2\|w\|_2^2.$$

Hence,

$$\mathbf{Var}_{x \sim \mu}[w \cdot x] \geq \mathbf{Var}_{x \sim U}[w \cdot x] - 3\max\{\epsilon_2, \epsilon_1^2, \epsilon_1 D\}\|w\|_2^2.$$

$\square$

Our next goal is to relate $\phi(\cdot; \rho^\star w)$ with the uniform measure $\phi(\cdot; 0)$. According to the above general lemma, we have to relate the first and second moments of $\phi(\cdot; \rho^\star w)$ with the ones of the uniform distribution $U = \phi(\cdot; 0)$.

*The Proof of Proposition 8.* Our goal is to apply Lemma 8. First, let us set

$$f_v(\rho) = \mathbf{E}_{x \sim \phi(\cdot; \rho w)}[v \cdot x],$$

for any unit vector $v \in \Theta$. Then it holds that

$$\left\|\mathbf{E}_{x \sim \phi(\cdot; 0)}[x] - \mathbf{E}_{x \sim \phi(\cdot; \rho^\star w)}[x]\right\|_2 = \sup_{v: \|v\|_2 = 1} |f_v(0) - f_v(\rho^\star)|.$$

Using the mean value theorem in $[0, \rho^\star]$ for any unit vector $v$, we have that there exists a $\xi = \xi_v \in (0, \rho^\star)$ such that

$$|f_v(0) - f_v(\rho^\star)| = \rho^\star |f_v'(\xi)|.$$

It suffices to upper bound $f_v'(\xi)$ for any unit vector $v$ and $\xi \in (0, \rho^\star)$. Let us compute $f_v'$. We have that

$$\frac{df_v}{d\rho} = \int_S (v \cdot x)\frac{d}{d\rho}\frac{e^{\rho(wx)}}{\int_S e^{\rho(wy)}dy}dx = \mathbf{E}_{x \sim \phi(\cdot; \rho w)}[(v \cdot x)(w \cdot x)] - \mathbf{E}_{x \sim \phi(\cdot; \rho w)}[v \cdot x]\mathbf{E}_{x \sim \phi(\cdot; \rho w)}[w \cdot x].$$

Since $x \in \mathcal{X}$, we have that
$$\sup_{v:\|v\|_2=1} \sup_{\xi\in(0,\rho^\star)} |f_v'(\xi)| \leq 2\|w\|_2 D^2 \,.$$
This gives that
$$\left\| \mathbf{E}_{x\sim\phi(\cdot;0)}[x] - \mathbf{E}_{x\sim\phi(\cdot;\rho^\star w)}[x] \right\|_2 \leq 2\rho^\star \|w\|_2 D^2 \,.$$
We then continue with controlling the second moment: it suffices to find $\epsilon_2$ such that for any $v \in \Theta$, it holds
$$v^\top \mathbf{E}_{x\sim\phi(\cdot;\rho^\star w)}[xx^\top]v \geq v^\top \mathbf{E}_{x\sim\phi(\cdot;0)}[xx^\top]v - \epsilon_2\|v\|_2^2 \,.$$
Let us set $g_v(\rho) = \mathbf{E}_{x\sim\phi(\cdot;\rho w)}[(v \cdot x)^2]$ for any vector $v \in \Theta$. We have that
$$|g_v(0) - g_v(\rho^\star)| = \rho^\star |g_v'(\xi)| \,,$$
where $\xi \in (0,\rho^\star)$. It holds that
$$\left| \frac{dg_v}{d\rho} \right| = \left| \mathbf{E}_{x\sim\phi(\cdot;\rho w)}[(v \cdot x)^2 (w \cdot x)] - \mathbf{E}_{x\sim\phi(\cdot;\rho w)}[(v \cdot x)^2] \, \mathbf{E}_{x\sim\phi(\cdot;\rho w)}[w \cdot x] \right| \leq 2\|v\|^2 \|w\|_2 D^3 \,.$$
This gives that for any $v \in \Theta$, it holds
$$v^\top \mathbf{E}_{x\sim\phi(\cdot;\rho^\star w)}[xx^\top]v \geq v^\top \mathbf{E}_{x\sim\phi(\cdot;0)}[xx^\top]v - 2\rho^\star \|w\|_2 D^3 \|v\|_2^2 \,.$$
Note that the above holds for $v = c$ too. [Lemma 8](#) gives us that
$$\mathbf{Var}_{x\sim\phi(\cdot;\rho^\star w)}[c \cdot x] \geq \mathbf{Var}_{x\sim\phi(\cdot;0)}[c \cdot x] - 3\max\{\epsilon_2, \epsilon_1^2, \epsilon_1 D\}\|c\|_2^2 \,,$$
where $\epsilon_1 = 2\rho^\star BD^2$ and $\epsilon_2 = 2\rho^\star BD^3$. This implies that by picking
$$\rho^\star = C_0 \frac{\alpha}{BD^3}$$
for some universal constant $C_0$, we get that
$$\mathbf{Var}_{x\sim\phi(\cdot;\rho^\star w)}[c \cdot x] = \Omega(\alpha\|c\|_2^2) \,.$$
$\square$

In this section, we considered $\rho^\star$ as indicated by the above [Proposition 8](#).

# G  Applications to Combinatorial Problems

In this section we provide a series of combinatorial applications of our theoretical framework ([Theorem 1](#)). In particular, for each one of the following combinatorial problems (that provably satisfy [Assumption 1](#)), it suffices to specify the feature mappings $\psi_S, \psi_\mathcal{I}$ and compute the parameters $C, D_S, D_\mathcal{I}, \alpha$.

## G.1  Maximum Cut, Maximum Flow and Max-$k$-CSPs

We first provide a general lemma for the variance of "linear tensors" under the uniform measure.

**Lemma 9** (Variance Lower Bound Under Uniform). *Let $n, k \in \mathbb{N}$. For any $w \in \mathbb{R}^{\binom{n}{k}}$, it holds that*
$$\mathbf{Var}_{x\sim U(\{-1,1\}^n)}[w \cdot x^{\otimes k}] = \sum_{\emptyset \neq S \subseteq [n]:|S|\leq k} w_S^2 \,.$$

*Proof.* For any $w \in \mathbb{R}^{\binom{n}{k}}$, it holds that
$$\mathbf{Var}_{x\sim U(\{-1,1\}^n)}[w \cdot x^{\otimes k}] = \mathbf{E}_{x\sim U(\{-1,1\}^n)}\left[(w \cdot x^{\otimes k})^2\right] - \mathbf{E}_{x\sim U(\{-1,1\}^n)}[w \cdot x^{\otimes k}]^2 \,.$$

Note that $w \in \mathbb{R}^{\binom{n}{k}}$ can be written as $w = (w_\emptyset, w_{-\emptyset})$ where $w_\emptyset$ corresponds to the constant term of the Fourier expansion and $w_{-\emptyset} = (w_S)_{\emptyset \neq S \subseteq [n]:|S|\leq k}$ is the vector of the remaining coordinates. The Fourier expansion implies that
$$\mathbf{Var}_{x\sim U(\{-1,1\}^n)}[w \cdot x^{\otimes k}] = \|w_{-\emptyset}\|_2^2 \,,$$
which yields the desired equality for the variance. $\square$

### G.1.1 Maximum Cut

Let us consider a graph with $n$ nodes and weighted adjacency matrix $A$ with non-negative weights. Maximum cut is naturally associated with the Ising model and, intuitively, our approach does not yield an efficient algorithm for solving Max-Cut since we cannot efficiently sample from the Ising model in general. To provide some further intuition, consider a single-parameter Ising model for $G = (V, E)$ with Hamiltonian $H_G(x) = \sum_{(i,j) \in E} \frac{1+x_i x_j}{2}$. Then the partition function is equal to $Z_G(\beta) = \sum_{x \in \{-1,1\}^V} \exp(\beta H_G(x))$. Note that when $\beta > 0$, the Gibbs measure favours configurations with alligned spins (ferromagnetic case) and when $\beta < 0$, the measure favours configurations with opposite spins (anti-ferromagnetic case). The antiferromagnetic Ising model appears to be more challenging. According to physicists the main reason is that its Boltzmann distribution is prone to a complicated type of long-range correlation known as 'replica symmetry breaking' [COLMS22]. From the TCS viewpoint, observe that as $\beta$ goes to $-\infty$, the mass of the Gibbs distribution shifts to spin configurations with more edges joining vertices with opposite spins and concentrates on the maximum cuts of the graph. Hence, being able to efficiently approximate the log-partition function for general Ising models, would lead to solving the Max-Cut problem.

**Theorem 2** (Max-Cut has a Compressed and Efficiently Optimizable Solution Generator). *Consider a prior over Max-Cut instances with $n$ nodes. For any $\epsilon > 0$, there exists a solution generator $\mathcal{P} = \{p(w) : w \in \mathcal{W}\}$ such that $\mathcal{P}$ is complete, compressed with description $\mathrm{poly}(n)\mathrm{polylog}(1/\epsilon)$ and $\mathcal{L} + \lambda R : \mathcal{W} \mapsto \mathbb{R}$ is efficiently optimizable via projected stochastic gradient descent in $\mathrm{poly}(n, 1/\epsilon)$ steps for some $\lambda > 0$.*

*Proof of Theorem 2.* It suffices to show that Max-Cut satisfies Assumption 1. Consider an input graph $G$ with $n$ nodes and Laplacian matrix $L_G$. Then

$$\texttt{MAXCUT} = \frac{1}{4} \max_{s \in \{-1,1\}^n} s^\top L_G s = \frac{1}{4} \min_{s \in \{-1,1\}^n} -s^\top L_G s \,.$$

We show that there exist feature mappings so that the cost of every solution $s$ under any instance/graph $G$ is a bilinear function of the feature vectors (cf. Item 2 of Assumption 1). We consider the correlation-based feature mapping $\psi_S(s) = (ss^\top)^\flat \in \mathbb{R}^{n^2}$, where by $(\cdot)^\flat$ we denote the vectorization/flattening operation and the negative Laplacian for the instance (graph), $\psi_{\mathcal{I}}(G) = (-L_G)^\flat \in \mathbb{R}^{n^2}$. Then simply setting the matrix $M$ to be the identity $I \in \mathbb{R}^{n^2 \times n^2}$ the cost of any solution $s$ can be expressed as the bilinear function $\psi_{\mathcal{I}}(G)^\top M \psi_S(s) = (-L_G^\flat)^\top (ss^T)^\flat = -s^\top L_G s$. We observe that (for unweighted graphs) with $n$ nodes the bit-complexity of the family of all instances $\mathcal{I}$ is roughly $O(n^2)$, and therefore the dimensions of the $\psi_S, \psi_{\mathcal{I}}$ feature mappings are clearly polynomial in the bit-complexity of $\mathcal{I}$. Moreover, considering unweighted graphs, it holds $\|\psi_{\mathcal{I}}(G)\|_2, \|\psi_S(s)\|_2, \|M\|_F \leq \mathrm{poly}(n)$. Therefore, the constants $D_S, D_{\mathcal{I}}, C$ are polynomial in the bit-complexity of the instance family.

It remains to show that our solution feature mapping satisfy the variance preservation assumption. For any $v$, we have that $\mathbf{Var}_{s \sim U(S)}[v \cdot \psi_S(s)] = \mathbf{Var}_{s \sim U(\{-1,1\}^n)}[v \cdot (ss^\top)^\flat] = \Omega(\|v\|_2^2)$, using Lemma 9 with $k = 2$, since $c_\emptyset = 0$ with loss of generality. $\qquad\square$

### G.1.2 Minimum Cut/Maximum Flow

Let us again consider a graph with $n$ nodes and Laplacian matrix $L_G$. It is known that the minimum cut problem is solvable in polynomial time when all the weights are positive. From the discussion of the maximum cut case, we can intuitively relate minimum cut with positive weights to the ferromagnetic Ising setting [dPS97]. We remark that we can consider the ferromagnetic parameter space $\mathcal{W}_{\mathrm{fer}} = \mathbb{R}_{\geq 0}^{\binom{n}{2}}$ and get the variance lower bound from Lemma 9. We constraint projected SGD in $\mathcal{W}_{\mathrm{fer}}$. This means that during any step of SGD our algorithm has to sample from a mixture of ferromagnetic models with known mixture weights. The state of the art approximate sampling algorithm from ferromagnetic Ising models achieves the following performance, improving on prior work [JS93, LSS19, CLV22, CGG$^+$19].

**Proposition 9** (Theorem 1.1 of [CZ22]). *Let $\delta_\beta, \delta_\lambda \in (0, 1)$ be constants and $\mu$ be the Gibbs distribution of the ferromagnetic Ising model specified by graph $G = (V, E), |V| = n, |E| = m$, parameters $\beta \in [1 + \delta_\beta, +\infty)^m$ and external field $\lambda \in [0, 1 - \delta_\lambda]^n$. There exists an algorithm that*

*samples $X$ satisfying* $\mathrm{TV}(X, \mu) \leq \epsilon$ *for any given parameter* $\epsilon \in (0, 1)$ *within running time*

$$m \left( \frac{\log n}{\epsilon} \right)^{O_{\delta_\beta, \delta_\lambda}(1)}.$$

This algorithm can handle general instances and it only takes a near-linear running time when parameters are bounded away from the all-ones vector. Our goal is to sample from a mixture of two such ferromagnetic Ising models which can be done efficiently. For simplicity, we next restrict ourselves to the unweighted case.

**Theorem 3** (Min-Cut has a Compressed, Efficiently Optimizable and Samplable Solution Generator)**.**
*Consider a prior over Min-Cut instances with $n$ nodes. For any $\epsilon > 0$, there exists a solution generator $\mathcal{P} = \{p(w) : w \in \mathcal{W}\}$ such that $\mathcal{P}$ is complete, compressed with description $\mathrm{poly}(n)\mathrm{polylog}(1/\epsilon)$, $\mathcal{L} + \lambda R : \mathcal{W} \mapsto \mathbb{R}$ is efficiently optimizable via projected stochastic gradient descent in $\mathrm{poly}(n, 1/\epsilon)$ steps for some $\lambda > 0$ and efficiently samplable in $\mathrm{poly}(n, 1/\epsilon)$ steps.*

*Proof.* We have that

$$\texttt{MINCUT} = \frac{1}{4} \min_{x \in \{-1, 1\}^n} x^\top L_G x \,.$$

The analysis (i.e., the selection of the feature mappings) is similar to the one of Theorem 2 with the sole difference that the parameter space is constrained to be $\mathcal{W}_{\mathrm{fer}}$ and $\psi_{\mathcal{I}}(G) = (L_G)^\flat$. We note that Proposition 9 is applicable during the optimization steps. Having an efficient approximate sampler for solutions of Min-Cut, it holds that the runtime of the projected SGD algorithm is $\mathrm{poly}(n, 1/\epsilon)$. We note that during the execution of the algorithm we do not have access to perfectly unbiased samples from mixture of ferromagnetic Ising models. However, we remark that SGD is robust to that inaccuracy in the stochastic oracle. For further details, we refer e.g., to [d'A08]). $\qquad\square$

### G.1.3 Max-$k$-CSPs

In this problem, we are given a set of variables $\{x_u\}_{u \in \mathcal{U}}$ where $|\mathcal{U}| = n$ and a set of Boolean predicates $P$. Each variable $x_u$ takes values in $\{-1, 1\}$. Each predicate depends on at most $k$ variables. For instance, Max-Cut is a Max-2-CSP. Our goal is to assign values to variables so as to maximize the number of satisfied constraints (i.e., predicates equal to 1). Let us fix a predicate $h \in P$, i.e., a Boolean function $h : \{-1, 1\}^n \to \{0, 1\}$ which is a $k$-junta. Using standard Fourier analysis, the number of satisfied predicates for the assignment $x \in \{-1, 1\}^n$ is

$$F(x) = \sum_{j=1}^{|P|} \sum_{S \subseteq [n], |S| \leq k} \widehat{h}_j(S) \prod_{u \in S} x_u \,,$$

where $\widehat{h}_j(S)$ is the Fourier coefficient of the predicate $h_j$ at $S$.

**Theorem 4** (Max-$k$-CSPs have a Compressed and Efficiently Optimizable Solution Generator)**.**
*Consider a prior over Max-k-CSP instances with $n$ variables, where $k \in \mathbb{N}$ can be considered constant compared to $n$. For any $\epsilon > 0$, there exists a solution generator $\mathcal{P} = \{p(w) : w \in \mathcal{W}\}$ such that $\mathcal{P}$ is complete, compressed with description $O(n^k)\mathrm{polylog}(1/\epsilon)$ and $\mathcal{L} + \lambda R : \mathcal{W} \mapsto \mathbb{R}$ is efficiently optimizable via projected stochastic gradient descent in $\mathrm{poly}(n^k, 1/\epsilon)$ steps for some $\lambda > 0$.*

*Proof.* Any instance of Max-$k$-CSP is a list of predicates (i.e., Boolean functions) and our goal is to maximize the number of satisfied predicated with a single assignment $s \in \{-1, 1\}^n$. We show that there exist feature mappings so that the cost of every solution $s$ under any instance/predicates list $P$ is a bilinear function of the feature vectors (cf. Item 2 of Assumption 1). We consider the order $k$ correlation-based feature mappings $\psi_S(s) = (s^{\otimes k})^\flat \in \mathbb{R}^{n^k}$, where by $(\cdot)^\flat$ we denote the flattening operation of the order $k$ tensor, and, $\psi_{\mathcal{I}}(P) = \psi_{\mathcal{I}}(h_1, \ldots, h_{|P|}) = -\sum_{j=1}^{|P|} ((\widehat{h}_j(S))_{S \subseteq [n], |S| \leq k})^\top \in \mathbb{R}^{n^k}$, where $(\widehat{h}_j(S))_{S \subseteq [n], |S| \leq k}$ is a vector of size $n^k$ with the Fourier coeffients of the $j$-th predicate. We take $\psi_{\mathcal{I}}$ being the coordinate-wise sum of these coefficients. The setting the matrix $M$ to be the identity matrix $I \in \mathbb{R}^{n^k \times n^k}$, we get that the cost of any solution $s$ can be expressed as the bilinear function $\psi_{\mathcal{I}}(P)^\top M \psi_S(s) =$

$-\sum_{j=1}^{|P|}\sum_{S\subseteq[n],|S|\leq k}\widehat{h}_j(S)\prod_{u\in S}x_u$. For any $h:\{-1,1\}^n\to\{0,1\}$, we get that the description size of any $\widehat{h}(S)$ is $\text{poly}(n,k)$ and so the dimensions of the $\psi_S,\psi_{\mathcal{I}}$ feature mappings are polynomial in the description size of $\mathcal{I}$. Moreover, we get that $\|\psi_{\mathcal{I}}(P)\|,\|\psi_S(s)\|,\|M\|\leq\text{poly}(n^k)$. Hence, the constants $D_S,D_{\mathcal{I}},C$ are polynomial in the description size of the instance family. Finally, we have that for any $v$, $\mathbf{Var}_{s\sim U(S)}[v\cdot\psi_S(s)]=\mathbf{Var}_{s\sim U(\{-1,1\}^n)}[v\cdot s^{\otimes k}]=\Omega(\|v\|_2^2)$, using Lemma 9, assuming that $v_\emptyset$ is 0 without loss of generality. This implies the result. $\qquad\square$

## G.2 Bipartite Matching and TSP

### G.2.1 Maximum Weight Bipartite Matching

In Maximum Weight Bipartite Matching (MWBM) there exists a complete bipartite graph $(A,B)$ with $|A|=|B|=n$ (the assumptions that the graph is complete and balanced is without loss of generality) with weight matrix $W$ where $W(i,j)$ indicates the value of the edge $(i,j),i\in A,j\in B$ and the goal is to match the vertices in order to maximize the value. Hence the goal is to maximize $L(\Pi)=W\cdot\Pi$ over all permutation matrices. By the structure of the problem some maximum weight matching is a perfect matching. Furthermore, by negating the weights of the edges we can state the problem as the following minimization problem: given a bipartite graph $(A,B)$ and weight matrix $W\in(\mathbb{R}\cup\{\infty\})^{n\times n}$, find a perfect matching $M$ with minimum weight. One of the fundamental results in combinatorial optimization is the polynomial-time blossom algorithm for computing minimum-weight perfect matchings by [Edm65].

We begin this section by showing a variance lower bound under the uniform distribution over the permutation group.

**Lemma 10** (Variance Lower Bound). *Let $U(\mathbb{S}_n)$ be the uniform distribution over $n\times n$ permutation matrices. For any matrix $W\in\mathbb{R}^{n\times n}$, with $\sum_i W_{ij}=0$ and $\sum_j W_{ij}=0$ we have*

$$\mathbf{Var}_{\Pi\sim U(\mathbb{S}_n)}[W\cdot\Pi]=\frac{\|W\|_{\mathrm{F}}^2}{n-1}.$$

*Proof.* We have that $\mathbf{E}_{\Pi\sim U(\mathbb{S}_n)}[\Pi_{ij}]=1/n$ and $\mathbf{E}_{\Pi\sim U(\mathbb{S}_n)}[\Pi_{ij}\Pi_{ab}]=\frac{\mathbb{1}\{i\neq a\wedge j\neq b\}}{n(n-1)}+\frac{\mathbb{1}\{i=a,j=b\}}{n}$. We have

$$\mathbf{Var}_{\Pi\sim U(\mathbb{S}_n)}[W\cdot\Pi]=\mathop{\mathbf{E}}_{\Pi\sim U(\mathbb{S}_n)}[(W\cdot\Pi)^2]-\left(\mathop{\mathbf{E}}_{\Pi\sim U(\mathbb{S}_n)}[W\cdot\Pi]\right)^2$$

$$=\sum_{i,j,a,b}W_{ab}W_{ij}\left(\frac{\mathbb{1}\{i\neq a,j\neq b\}}{n(n-1)}+\frac{\mathbb{1}\{i=a,j=b\}}{n}\right)-\left(\sum_{i,j}\frac{W_{ij}}{n}\right)^2$$

$$=\frac{1}{n}\sum_{i,j}W_{ij}^2+\sum_{i,j,a,b}W_{ij}W_{ab}\frac{\mathbb{1}\{i\neq a,j\neq b\}}{n(n-1)}$$

$$=\frac{\|W\|_{\mathrm{F}}^2}{n}+\sum_{i,j,a,b}W_{ij}W_{ab}\frac{\mathbb{1}\{i\neq a,j\neq b\}}{n(n-1)},$$

where to obtain the third equality we used our assumption that $\sum_{ij}W_{ij}=0$. We observe that, by our assumption that $\sum_b W_{ab}=0$ for all $a$ it holds $\sum_{b\neq j}W_{ab}=-W_{aj}$ and therefore, we have

$$\sum_b W_{ab}\mathbb{1}\{i\neq a,j\neq b\}=\mathbb{1}\{i\neq a\}\sum_b W_{ab}\mathbb{1}\{j\neq b\}=-\mathbb{1}\{i\neq a\}W_{aj}.$$

Similarly, using the fact that $\sum_{a\neq i}W_{aj}=-W_{ij}$ we obtain that

$$\sum_{ab}W_{ab}\mathbb{1}\{i\neq a,j\neq b\}=\sum_a -W_{aj}\mathbb{1}\{i\neq a\}=W_{ij}.$$

Therefore, using the above identity, we have that

$$\sum_{i,j,a,b}W_{ij}W_{ab}\frac{\mathbb{1}\{i\neq a,j\neq b\}}{n(n-1)}=\sum_{i,j}\frac{W_{ij}^2}{n(n-1)}=\frac{\|W\|_{\mathrm{F}}^2}{n(n-1)}.$$

Combining the above we obtain the claimed identity. $\qquad\square$

**Remark 8.** *We note that in MWBM the conditions $\sum_i W_{ij} = 0$ and $\sum_j W_{ij} = 0$ are without loss of generality.*

We next claim that there exists an efficient algorithm for (approximately) sampling such permutation matrices.

**Lemma 11** (Efficient Sampling). *There exists an algorithm that generates approximate samples from the Gibbs distribution $p(\cdot; W)$ with parameter $W$ over the symmetric group, i.e., $p(\Pi; W) \propto \exp(W \cdot \Pi)1\{\Pi \in \mathbb{S}_n\}$, in $\mathrm{poly}(n)$ time.*

*Proof.* This lemma essentially requires approximating the permanent of a weighted matrix, since this would imply that one has an approximation of the partition function. Essentially, our goal is to generate a random variable $X$ that is $\epsilon$-close in statistical distance to the probability measure

$$p(\Pi; W) \propto \exp(W \cdot \Pi)1\{\Pi \in \mathbb{S}_n\}\,.$$

Note that the partition function is

$$Z(W) = \sum_{\Pi \in \mathbb{S}_n} e^{W \cdot \Pi} = \sum_{\Pi \in \mathbb{S}_n} \prod_{(i,j)} e^{W_{ij}\Pi_{ij}} = \sum_{\sigma \in \mathbb{S}_n} \prod_{i \in [n]} A_{i,\sigma(i)}\,,$$

where $A$ is a non-negative real matrix with entries $A_{ij} = \exp(W_{ij})$. Note that in the third equality, we used the isomorphism between permutations and permutation matrices. Hence, $Z(W)$ is exactly the permanent of the matrix $A$.

**Proposition 10** ([JSV04]). *There exists a fully polynomial randomized approximation scheme for the permanent of an arbitrary $n \times n$ matrix $A$ with non-negative entries.*

To conclude the proof of the lemma, we need the following standard result.

**Proposition 11** (See Appendix H and [Sin12, Jer03]). *For self-reducible problems, fully polynomial approximate integration and fully polynomial approximate sampling are equivalent.*

This concludes the proof since weighted matchings are self-reducible (see Appendix H). $\square$

The above lemma establishes our goal:

**Theorem 5** (MWBM has a Compressed, Efficiently Optimizable and Samplable Solution Generator). *Consider a prior over MWBM instances with $n$ nodes. For any $\epsilon > 0$, there exists a solution generator $\mathcal{P} = \{p(w) : w \in \mathcal{W}\}$ such that $\mathcal{P}$ is complete, compressed with description $\mathrm{poly}(n)\mathrm{polylog}(1/\epsilon), \mathcal{L} + \lambda R : \mathcal{W} \mapsto \mathbb{R}$ is efficiently optimizable via projected stochastic gradient descent in $\mathrm{poly}(n, 1/\epsilon)$ steps for some $\lambda > 0$ and efficiently samplable in $\mathrm{poly}(n, 1/\epsilon)$ steps.*

*Proof.* Consider an input graph $G$ with $n$ nodes and adjacency matrix $E$. The feature vector corresponding to a matching can be represented as a binary matrix $\Pi \in \{0, 1\}^{n \times n}$ with $\sum_j \Pi_{ij} = 1$ for all $i$ and $\sum_i \Pi_{ij} = 1$ for all $j$, i.e., $\Pi$ is a permutation matrix. Then

$$\mathtt{MWBM} = \max_{\Pi \in \mathbb{S}_n} E \cdot \Pi = \min_{\Pi \in \mathbb{S}_n} -E \cdot \Pi\,.$$

Therefore, for a candidate matching $s$, we set $\psi_S(s)$ to be the matrix $\Pi$ defined above. Moreover, the feature vector of the graph is the negative (flattened) adjacency matrix $-E^\flat$. The cost oracle is then $L(R; E) = -\sum_{ij} E_{ij}M_{ij}R_{ij}$ perhaps for an unknown weight matrix $M_{ij}$ (see Remark 6). This means that the dimensions of the feature mappings $\psi_S, \psi_\mathcal{I}$ are polynomial in the bit complexity of $\mathcal{I}$. Moreover, we get that $\|\psi_\mathcal{I}(I)\|_\mathrm{F}, \|\psi_S(s)\|_\mathrm{F}, \|M\|_F \leq \mathrm{poly}(n)$. We can employ Lemma 10 to get the variance lower bound under the uniform probability distribution in the subspace induced by the matrices satisfying Lemma 10, i.e., the matrices of the parameter space (see Remark 9). Finally, (approximate) sampling from our solution generators can be done efficiently using Lemma 11 and hence (noisy) projected SGD will have a runtime of order $\mathrm{poly}(n, 1/\epsilon)$ (as in the case of Min-Cut). $\square$

We close this section with a remark about Item 3 of Assumption 1.

**Remark 9.** *We note that Item 3 of [Assumption 1](#) can be weakened. We use our variance lower bound in order to handle inner products of the form $w \cdot x$ where $w$ will lie in the parameter space and $x$ is the featurization of a solution that lies in some space $X$. Hence it is possible that $w$ lies in a low-dimensional subspace of $X$. For our optimization purposes, it suffices to provide variance lower bounds only in the subspace where $w$ lies into.*

### G.2.2 Travelling Salesman Problem

Let us consider a weighted clique $K_n$ with $n$ vertices and weight matrix $W \in \mathbb{R}^{n \times n}$. A solution to the TSP instance $W$ is a sequence $\pi : [n] \to [n]$ of the $n$ elements indicating the TSP tour $(\pi(1), \pi(2), \ldots, \pi(n), \pi(1))$ and suffers a cost

$$L(\pi) = \sum_{i=1}^{n-1} W_{\pi(i),\pi(i+1)} + W_{\pi(n),\pi(1)}.$$

Crucially, the allowed sequences are a proper subset of all possible permutations. For instance, the permutations with fixed points or small cycles are not allowed. In particular, the solution space of TSP corresponds to the set of cyclic permutations with no trivial cycles, i.e., containing an $n$-cycle. Clearly, the number of $n$-cycles is $(n-1)!$. The goal is to find a tour of minimum cost. Our first goal is to write the cost objective as a linear function of the weight matrix $W$ and the feasible solutions, which correspond to cyclic permutations. To this end, we can think of each cyclic permutation $\pi$ as a cyclic permutation matrix $\Pi \in \{0,1\}^{n \times n}$. Then, the desired linearization is given by $L(\Pi) = W \cdot \Pi$ (for a fixed graph instance).

Our next task is to provide a Gibbs measure that generates random cyclic permutations. Let $\mathbb{C}_n$ be the space of $n \times n$ cyclic permutation matrices. Then we have that the tour $\Pi$ is drawn from

$$p_W(\Pi) = \frac{\exp(W \cdot \Pi) \mathbb{1}\{\Pi \in \mathbb{C}_n\}}{\sum_{\Pi' \in \mathbb{C}_n} \exp(W \cdot \Pi')},$$

where $W$ is the weight matrix. The following key lemma provides guarantees for the performance of our approach to TSP. This lemma allows us to show that the number of optimization steps is $\mathrm{poly}(n, 1/\epsilon)$.

**Lemma 12** (Variance Lower Bound). *Let $U(\mathbb{C}_n)$ be the uniform distribution over $n \times n$ cyclic permutation matrices. For any matrix $W \in \mathbb{R}^{n \times n}$, with $\sum_i W_{ij} = 0$ and $\sum_j W_{ij} = 0$ we have*

$$\mathbf{Var}_{\Pi \sim U(\mathbb{C}_n)}[W \cdot \Pi] \geq \frac{\|W\|_{\mathrm{F}}^2}{(n-1)(n-2)}.$$

*Proof.* The first step is to compute some standard statistics about cyclic permutations (see [Lemma 13](#)). [Lemma 13](#) and the analysis of [Lemma 10](#) gives us that $\mathbf{Var}_{\Pi \sim U(\mathbb{C}_n)}[W \cdot \Pi]$ is equal to

$$\frac{\|W\|_{\mathrm{F}}^2}{n-1} + \sum_{i,j,a,b} W_{ij} W_{ab} \left( \frac{\mathbb{1}\{i \neq a = j \neq b\}}{(n-1)(n-2)} + \frac{\mathbb{1}\{j \neq b = i \neq a\}}{(n-1)(n-2)} + \frac{\mathbb{1}\{i \neq a \neq j \neq b \neq i\}}{(n-1)(n-3)} \right).$$

Let us set

$$A_1 = \sum_{i,j,a,b} W_{ij} W_{ab} \mathbb{1}\{i \neq a = j \neq b\}.$$

We have that

$$\sum_b W_{ab} \mathbb{1}\{i \neq a\} \mathbb{1}\{a = j\} \mathbb{1}\{j \neq b\} = \mathbb{1}\{i \neq a\} \mathbb{1}\{a = j\} \sum_b W_{ab} \mathbb{1}\{j \neq b\} = -\mathbb{1}\{i \neq a\} \mathbb{1}\{a = j\} W_{aj}.$$

Hence

$$A_1 = -\sum_{i,j,a} W_{ij} W_{aj} \mathbb{1}\{i \neq a\} \mathbb{1}\{a = j\} = -\sum_{i,j} W_{ij} W_{jj} \mathbb{1}\{i \neq j\} = \sum_j W_{jj}^2.$$

Due to symmetry, $A_2 = \sum_{i,j,a,b} W_{ij} W_{ab} \mathbb{1}\{j \neq b = i \neq a\} = \sum_j W_{jj}^2$. It remains to argue about

$$A_3 = \sum_{i,j,a,b} W_{ij} W_{ab} \mathbb{1}\{i \neq a \neq j \neq b \neq i\}.$$

We have that

$$\sum_b W_{ab} 1\{i \neq a\} 1\{a \neq j\} 1\{j \neq b\} 1\{b \neq i\} = -1\{i \neq a\} 1\{a \neq j\}(W_{aj} + W_{ai}).$$

This gives that

$$A_3 = -\sum_{i,j,a} W_{ij}(W_{aj} + W_{ai}) 1\{i \neq a\} 1\{a \neq j\}.$$

Note that

$$\sum_{i,j,a} W_{ij} W_{aj} 1\{i \neq a\} 1\{a \neq j\} = \sum_{j,a} W_{aj} 1\{a \neq j\} \sum_i W_{ij} 1\{i \neq a\} = -\sum_{j,a} W_{aj}^2 1\{a \neq j\}.$$

This implies that

$$A_3 = \sum_{j \neq a} W_{aj}^2 + \sum_{i \neq a} W_{ai}^2 = 2\sum_{j \neq a} W_{aj}^2.$$

In total, this gives that

$$\mathbf{Var}_{\Pi \sim U(\mathbb{C}_n)}[W \cdot \Pi] = \frac{\|W\|_{\mathrm{F}}^2}{n-1} + \frac{2\sum_j W_{jj}^2}{(n-1)(n-2)} + \frac{2\sum_{i \neq j} W_{ij}^2}{(n-1)(n-3)} \geq \frac{\|W\|_{\mathrm{F}}^2}{n-1} + \frac{\|W\|_{\mathrm{F}}^2}{(n-1)(n-2)}.$$

$\square$

**Remark 10.** *We note that in TSP the conditions $\sum_i W_{ij} = 0$ and $\sum_j W_{ij} = 0$ are without loss of generality.*

The next lemma is a generic lemma that states some properties of random cyclic permutation matrices.

**Lemma 13.** *Consider a uniformly random cyclic permutation matrix $\Pi$. Let us fix $i \neq j$ and $a \neq b$. Then*

- $\mathbf{E}[\Pi_{ij}] = \frac{1}{n-1}$.

- $\mathbf{E}[\Pi_{ij}\Pi_{ab}] = \frac{1\{i \neq a = j \neq b\}}{(n-1)(n-2)} + \frac{1\{j \neq b = i \neq a\}}{(n-1)(n-2)} + \frac{1\{i \neq a \neq j \neq b \neq i\}}{(n-1)(n-3)} + \frac{1\{i=a, j=b\}}{n-1}$.

*Proof.* First, note that any matrix that corresponds to a cyclic permutation does not contain fixed points and so the diagonal elements are 0 deterministically. For the first item, the number of cyclic permutations such that $i \to j$ (i.e., $\Pi_{ij} = 1$) is $(n-2)!$. This implies that the desired expectation is $(n-2)!/|\mathbb{C}_n| = 1/(n-1)$. For the second item, if $i = a, j = b$, we recover the first item. Otherwise if $i = a$ or $j = b$, then the expectation vanishes since we deal with permutation matrices. Finally, let us consider the case where $i \neq a$ and $j \neq b$. Our goal is to count the number of cyclic permutations with $i \to j$ and $a \to b$.

- If $i \neq a \neq j \neq b \neq i$, then there are $n$ choices to place $i$ and $n-2$ choices to place $a$. Then there are $(n-4)!$ possible orderings for the remaining elements. This gives an expectation equal to $1/((n-1)(n-3))$.

- If $i \neq a = j \neq b$ or $j \neq b = i \neq a$, then there are $n$ choices for $i$ and $(n-3)!$ orderings for the remaining elements. Hence, the expectation is $1/((n-1)(n-2))$.

$\square$

We note that sampling from our solution generators is the reason that we cannot find an optimal TSP solution efficiently. In general, an algorithm that has converged to an almost optimal parameter $W^\star$ has to generate samples from the Gibbs measure that is concentrated on cycles with minimum weight. In this low-temperature regime, sampling is NP-hard. We are now ready to state our result.

**Theorem 6** (TSP has a Compressed, Efficiently Optimizable Solution Generator). *Consider a prior over TSP instances with $n$ nodes. For any $\epsilon > 0$, there exists a solution generator $\mathcal{P} = \{p(w) : w \in \mathcal{W}\}$ such that $\mathcal{P}$ is complete, compressed with description $\mathrm{poly}(n)\mathrm{polylog}(1/\epsilon)$ and $\mathcal{L} + \lambda R : \mathcal{W} \mapsto \mathbb{R}$ is efficiently optimizable via projected stochastic gradient descent in $\mathrm{poly}(n, 1/\epsilon)$ steps for some $\lambda > 0$.*

*Proof.* Consider an input graph $G$ with $n$ nodes and weighted adjacency matrix $E$. The feature vector is again a permutation matrix $\Pi$ with the additional constraint that $\Pi$ has to represent a single cycle (a tour over all cities). Then

$$\texttt{TSP} = \min_{\Pi \in \mathbb{C}_n} E \cdot \Pi \, .$$

The cost function for TSP is $L(\Pi; E) = \sum_{ij} E_{ij} M_{ij} \Pi_{ij}$. We refer to Theorem 5 for the details about the feature mappings. We can finally use Lemma 12 to obtain a variance lower bound (in the subspace induced by the parameters satisfying this lemma, see Remark 9) under the uniform measure over the space of cyclic permutation matrices $\mathbb{C}_n$. $\square$

# H    Sampling and Counting

In this section, we give a quick overview of the connections between approximate sampling and counting. For a formal treatment, we refer to [Sin12].

In what follows, $\sigma$ may be thought of as an encoding of an instance of some combinatorial problem, and the $\omega$ of interest are encodings of the structures we wish to generate. Consider a weight function $W$ and assume that $W(\sigma, \omega)$ is computable in time polynomial in $|\sigma|$.

**Definition 4** (Approximate Sampling). *A fully polynomial approximate sampler for $(\Omega_\sigma, \pi_\sigma)$ is a Probabilistic Turing Machine which, on inputs $\sigma$ and $\epsilon \in \mathbb{Q}_+$ $(0 < \epsilon \leq 1)$, outputs $\omega \in \Sigma^\star$, according to a measure $\mu_\sigma$ satisfying $\mathrm{TV}(\pi_\sigma, \mu_\sigma) \leq \epsilon$, in time bounded by a bivariate polynomial in $|\sigma|$ and $\log(1/\epsilon)$.*

One of the main applications of sampling is to approximate integration. In our setting this means estimating $Z(\sigma)$ to some specified relative error.

**Definition 5** (Approximate Integration). *A fully polynomial randomized approximation scheme for $Z(\sigma)$ is a Probabilistic Turing Machine which on input $\sigma, \epsilon$, outputs an estimate $\widehat{Z}$ so that*

$$\mathbf{Pr}[Z/(1+\epsilon) \leq \widehat{Z} \leq (1+\epsilon)Z] \geq 3/4 \, ,$$

*and which runs in time polynomial in $|\sigma|$ and $1/\epsilon$.*

**Definition 6** (Self-Reducible Problems). *An NP search problem is self-reducible if the set of solutions can be partitioned into polynomially many sets each of which is in a one-to-one correspondence with the set of solutions of a smaller instance of the problem, and the polynomial size set of smaller instances are efficiently computable.*

For instance, consider the relation `MATCH` which associates with an undirected graph $G$ all matchings (independent sets of edges) of $G$. Then `MATCH` is self-reducible since, for any edge $e = (u, v) \in E(G)$, we have that

$$\texttt{MATCH}(G) = \texttt{MATCH}(G_1) \cup \{M \cup \{e\} : M \in \texttt{MATCH}(G_2)\} \, ,$$

where $G_1$ is the graph obtained by deleting $e$ and $G_2$ is the graph obtained be deleting both $u$ and $v$ together with all their incident edges.

**Theorem 7** (See Corollary 3.16 in [Sin12]). *For self-reducible problems, approximate integration and good sampling are equivalent.*

We remark that the above result holds for the more general class of self-partitionable problems.

# I    Details of the Experimental Evaluation

We investigate the effect of the entropy regularizer (see Equation (2)) in a very simple setting: we try to find the Max-Cut of a fixed graph $G$, i.e., the support of the prior $\mathcal{R}$ is a single graph. We show that while the unregularized objective is often "stuck" at sub-optimal solutions – and this happens even for very small instances (15 nodes) – of the Max-Cut problem, the regularized version (with the fast/slow mixture scheme) is able to find the optimal solutions. We consider an instance randomly generated by the Erdős–Rényi model $G(n, p)$ and then optimize the "vanilla" loss $\mathcal{L}$ and the regularized loss $\mathcal{L}_\lambda$ defined in Equation (3). The solutions are vectors $s \in \{\pm 1\}^n$. We first use the feature mapping $\psi_S(s) = (ss^\top)^\flat$ described in Section 1.1 and an exponential family solution generator that samples a

```python
class FastSlowMixture(torch.nn.Module):
def __init__(self, dimension, rho):
    """
    The Model parameters.
    """
    super().__init__()

    self.l1 = torch.nn.Parameter(torch.empty(30, dimension))
    torch.nn.init.kaiming_uniform_(self.l1, a=5**0.5)

    self.l2 = torch.nn.Parameter(torch.empty(10, 30))
    torch.nn.init.kaiming_uniform_(self.l2, a=5**0.5)

    self.l3 = torch.nn.Parameter(torch.empty(1, 10))
    torch.nn.init.kaiming_uniform_(self.l3, a=5**0.5)

    self.a2 = torch.nn.ReLU()
    self.a1 = torch.nn.ReLU()

    self.rho = rho

def forward(self, x, is_cold=True):

    temp = self.rho * (1. - is_cold) + is_cold

    out = x
    out = torch.nn.functional.linear(out, temp * self.l1)
    out = self.a1(out)
    out = torch.nn.functional.linear(out, temp * self.l2)
    out = self.a2(out)
    out = torch.nn.functional.linear(out, temp * self.l3)

    return out
```

Figure 4: Our implementation of the fast/slow network. The output of the network is the log-density (score) of a solution $s \in \{\pm 1\}^{\text{dimension}}$. If evaluated with the is-cold set to False, the parameters of every linear layer are re-scaled by the inverse temperature rho.

solution $s$ with probability $\propto \exp(w \cdot \psi_S(s))$ for some weight vector $w \in \mathbb{R}^{n^2}$. We also consider optimizing a simple 3-layer ReLU network as solution generator with input $s \in \{\pm 1\}^n$ on the same random graphs. We generate 100 random $G(n, p)$ graphs with $n = 15$ nodes and $p = 0.5$ and train solution generators using both the "vanilla" and the entropy-regularized loss functions. We perform 600 iterations and, for the entropy regularization, we progressively decrease the regularization weight, starting from 10, and dividing it by 2 every 60 iterations. We used a fast/slow mixing with mixture probability 0.2 and inverse temperature rho=0.03 (see Figure 4).

For convenience, we present a pytorch implementation of our simple 3-layer ReLU network here. For more details we refer to our full code submitted in the supplementary material.

Out of the 100 trials we found that our proposed objective was always able to find the optimal cut while the model trained with the vanilla loss was able to find it for approximately $65\%$ of the graphs (for 65 out of 100 using the linear network and for 66 using the ReLU network). In Figure 2 we show two instances where the model trained with the "vanilla" loss gets stuck on a sub-optimal solution while the entropy-regularized one succeeds in finding the optimal solution. Our experiments show that the regularization term and the fast/slow mixture scheme that we introduced to achieve our main theoretical convergence result, see Section 3 and Proposition 4, are potentially useful for training more realistic models for bigger instances and we leave more extensive experimental evaluation as an interesting direction for future work.

We note that, similarly to our theoretical results, our sampler in this experimental section is of the form $e^{\text{score}(s;w)}$, where $s \in \{-1, 1\}^n$ ( here $n$ is the number of nodes in the graph) is a candidate solution of the Max-Cut problem. The function used is a 3-layer MLP (see Figure Figure 4). Since the instances that we consider here are small ($n = 15$) we can explicitly compute the density (score) of every solution and use that to compute the expected gradient. For larger instances, one could use some approximate sampler (e.g., via Langevin dynamics) to generate samples. The main message of the current experimental section is that even for very small instances of Max-Cut (i.e., with 15 nodes), optimizing the vanilla objective is not sufficient and the iteration gets trapped in local optima. In contrast, our entropy regularized always manages to find the optimal cut.

