# OpenReview forum: "Optimizing Solution-Samplers for Combinatorial Problems: The Landscape of Policy-Gradient Method"
_NeurIPS.cc/2023/Conference — NeurIPS 2023 oral_

### Official Review · Reviewer_dPn4 · 2023-07-06

**Soundness:** 3 good
**Presentation:** 4 excellent
**Contribution:** 4 excellent
**Rating:** 10
**Confidence:** 3

**Summary:**

The paper presents a thorough consideration on the parametrization of optimizable solution generators that allow using gradient descent to find solutions for combinatorial optimization methods. They find a set of assumptions that enable complete, compressed and efficiently optimizable representations, in particular existence of feature maps of solution and instances onto bounded (norm, diameter and dimensionality) and variance preserving feature spaces which allow for a bilinear cost oracle $c(I,S)=f_I^T M f_S$ for instance/solution features $f_I/f_S$ and a matrix $M$ s.t. $\Vert M \Vert_F\$ is bounded by a constant $C$.

The implications of this result (including the existence of such parametrizations for TSP) are discussed, notably that existance of such generators does not mean P=NP since sampling from the generator might still take exponential time.

The theory is developed on max/min cut, max-k-CSP, max-weight BP matching and TSP and experimentally evalauted on max-cut

**Strengths:**

Originality: I love this paper for taking the obvious in hindsight question for actually asking what types of representations make combinatorial optimisation via first order methods work on a *principles* basis
Clarity: The paper is superbly written and conveys it's ideas and limitations well
Quality: while I did not have the time to carefully check every brief, on a cursory reading they appear to make sense and align well with intuitions.
Significance: I think this paper will completely change how the deep learning community will think about dealing with combinatorial optimization (or at least I hope so)

**Weaknesses:**

I cannot point to any

**Questions:**

None

**Limitations:**

I think everything was addressed

---

> ### Author Rebuttal · Authors · 2023-08-08
>
>
> We are very encouraged from the very positive feedback of the reviewer and their excitement about our work and potential impact.  We are also very excited about our work and believe that our work will initiate a principled theoretical study of neural combinatorial optimization and its theoretical insights will be of practical significance.

---

> > ### Comment · Reviewer_dPn4 · 2023-08-12
> > **Acknowledging rebuttal**
> >
> > Commenting to confirm I have read the rebuttals and other reviews and will digest and engage in discussion.

---

### Official Review · Reviewer_m1vy · 2023-07-06

**Soundness:** 3 good
**Presentation:** 3 good
**Contribution:** 4 excellent
**Rating:** 6
**Confidence:** 4

**Summary:**

The paper provides a theoretical analysis of policy-gradient methods for combinatorial optimization problems. It defines a set of three desirable properties one might wish to be fulfilled for such a property, namely being able to generate an approximately optimal solution, being small in size, and enabling efficient optimization via SGD. The main result of the paper is to prove that there exists a policy gradient method satisfying all three properties simultaneously. The proposed method is general enough to capture many well-known combinatorial optimization problems. A small empirical study is provided, too.

**Strengths:**

I think the idea behind this work is great. Theoretical foundations for reinforcement learning methods applied to CO problems are highly needed. Defining a set of desirable properties and analyzing whether and how they can be satisfied simultaneously seems to be the right approach.

**Weaknesses:**

Since I believe that the contributions of this paper are great, I feel really sorry that I cannot give a more positive evaluation at the moment. I try to give as much constructive feedback as possible and encourage the authors to revise the manuscript and, if not accepted here, resubmit to another top venue. My main concern is:
- I have a few mathematical issues / questions, where I do not understand the paper. I think one source of these issues is that the model of computation is not well-defined. The authors wildly switch between bit-representations and real models of computation. To solve these issues, the authors should clearly define in which model of computation they work and make all assumptions / requirements / statements consistent with this model. There are a few more unrelated issues, see my more specific questions in the "Questions" section.

Apart from that, I have a whole bunch of secondary comments, see below:
- I think the paper (implicitly and sometimes explicitly) oversells the contribution of [BPL+16] to amplify the motivation for their own work. So far, reinforcement learning is NOT a state-of-the-art method for combinatorial optimization problems like TSP. The work of [BPL+16] is great as a proof of concept, but statements like "[BPL+16] [...] generate very good solutions for (Euclidean) TSP instances" without mentioning that these are very small toy problems should be avoided. I think the fact that NNs for CO are still in their beginnings should be pointed out more clearly and the motivation of this paper should build upon a larger variety of prior work than only [BPL+16].
- line 103: I suppose you want to refer to Def. 2 and not to Qu. 1 here (in particular, because otherwise you refer to Qu. 1 before even stating it).
- line 143: I find it more natural to write this with $L(s,I)$ instead of $\mathcal{O}(s,I)$. I understand that this is the same by definition, but here the focus is more on the cost structure and less on the fact that such an oracle exists. Maybe one could even drop the notation $\mathcal{O}(s,I)$ everywhere in the paper and use always $L$? In Definition 1, one could instead just write in words that such an oracle exists.
- line 196: I feel that the split into supervised and unsupervised models here is very artificial because papers in both categories apply sophisticated algorithms mixing very different paradigms. I doubt that the negative result [YGS20] is directly applicable to the settings of the three papers you cite for the supervised case. I suggest not to artificially split the related work into these two categories.
- Related work: even though of a very different flavor, I think the following two papers about the theoretical ability of neural networks to solve CO problems should be discussed in the related work section: Hertrich, C., & Skutella, M. (2023). Provably good solutions to the knapsack problem via neural networks of bounded size. INFORMS Journal on Computing. AND: Hertrich, C., & Sering, L. (2023). ReLU neural networks of polynomial size for exact maximum flow computation. In International Conference on Integer Programming and Combinatorial Optimization.
- line 219: please either provide a proof or a reference for the reformulation of the MaxCut problem via the Laplacian.
- line 241: say that Thm. 4 is in the appendix.
- line 246: a unknown -> an unknown. Also the whole sentence is a bit hard to read, maybe split into two?
- line 286: "the point $\bar{W}=-\tau M$ when $\tau\rightarrow+\infty$" is not a point. What you write here is not mathematically precise, even though I understand what you mean. Consider revising.
- lines 323/324: I recommend a comma between "fast" and "making".
- line 344: there is a "to" too much (between "we" and "pick").
- line 382: you should not assume that everyone knows what $G(n,p)$ is.

**Questions:**

- line 73: why is this the gradient? Please provide some explanation. In particular: where does the logarithm come from?
- line 75: what do you mean by "using only access to a solution cost oracle"? Do you want to point out that something particular is not used? If so, what? I suppose you also need to be able to sample $I$ and $s$ according to their respective distributions, and you need to be able to compute the gradient of $log(p(s;I;w))$ w.r.t. $w$.
- line 96: what is the "description size of the parameter space $\mathcal{W}$"? Is it the number of bits required to represent any parameter $w\in \mathcal{W}$? But then, $\mathcal{W}$ is a finite set, so how can you ever perform gradient descent on it?
- line 99: why do you allow running time polynomial in $1/\epsilon$, but the parameter space must be polynomial in $\log(1/\epsilon)$?
- line 107: I do not understand what "the full parametrization of all distributions over the hypercube" is. There are uncountably many such distributions, so how can you ever represent them in a set $\mathcal{W}$ of finite description size?
- Remark 5: Is this a formal statement or an intuitive statement? I really would like to see more details about how exactly the cited paper implies a negative result for neural network solution samplers in the setting of your paper.
- Remark 6: What if I WANT to encode the weights into the instances? I suppose you would it call "known" weights then. But it is less clear how to define the feature maps then and still preserve bilinearity. I do not see why what you call "unknown" weights is the more general / difficult case.

**Limitations:**

In principal, the authors provide all details required to understand the scope of their results. The discussion on limitations could be improved by answering the following two questions:
- Are there natural combinatorial optimization problems to which the framework is not applicable? If so, which problems, and why?
- It seems like the proposed method is theoretically superior to what people use in practice. Is this true? And if so, why is it not the standard method in practical settings nowadays? Do you expect it to become that?

---

> ### Author Rebuttal · Authors · 2023-08-08
>
> We would like to thank the reviewer (i) for finding the contribution of our paper great and for providing extensive constructive feedback, (ii) for pointing out various typos in the current draft, which we will fix in our first revision and (iii) for proposing additional references and suggestions for the Related Work section; we added the papers' for a more complete and detailed exposition. We continue by explicitly addressing the reviewer's questions.
>
> > I think the paper (implicitly and sometimes explicitly) oversells the contribution of [BPL+16] ...
>
> The work of  [BPL+16] is the (experimental) work that introduced the framework of Neural Combinatorial Optimization. Our work is the first theoretical approach to design a rigorous and principled way to study and argue about the fundamental questions behind this problem. Hence, we chose to design our theoretical framework on top of this (arguably) well-accepted work. We do not claim that the experimental results of  [BPL+16] are SOTA. However, we find that this work is the simplest and most natural setting to go from a completely practical to a rigorous theoretical framework. Hence, our work does not build on the experimental contribution of  [BPL+16] (as the reviewer correctly mentions there are various follow-up works that improve on this front) but builds on the conceptual contribution of  [BPL+16] providing the first formal theoretical guarantees. We will clarify the fact that the results of  [BPL+16]  are for smaller instances.
>
> > Line 219
>
> This is standard, see e.g., Spielman (2010). We will add a reference, as the Reviewer proposes.
>
> [Spielman, Algorithms, graph theory, and linear equations in Laplacian matrices, 2010]
>
> > Line 73
>
> This is the standard expression for policy gradient, for the reference see e.g., Section 2 and 5 in the paper of Kakade (2001) as we mention in our manuscript and for a proof see the book by Szepesvári (2022).
>
> [Kakade, A natural policy gradient, 2001]
>
> [Szepesvári, Algorithms for reinforcement learning, 2022]
>
> > Line 75
>
> One does not need access to an explicit representation of the cost function (e.g., the Laplacian matrix of the Graph in the Max-Cut problem) but only black box access to the cost oracle (i.e., the values of the function).
>
> > Model of Computation
> >
> > Line 96
>
> Gradient descent is not explicitly performed on a finite set – it is just the rounding of the iterates (due to finite memory in its implementation) that implicitly makes the set discrete.  Our description size captures exactly this finite precision issue. Say we require $d$ parameters for the parameter space of gradient descent; then a description size of $O(d \log(1/\epsilon))$ would mean that when you implement gradient descent you should use $O(d \log(1/\epsilon))$ bits for the representation of its iterates (roughly $\log(1/\epsilon)$ bits for each parameter).  Since the loss that we use is $\mathrm{poly}(d)$-Lipschitz with respect to the parameter $w$ (assuming that $w$ belongs in the continuous space), rounding the iterates of Gradient descent to $O(d\log(1/\epsilon))$ bits does not introduce substantial error. We performed the analysis of gradient descent in the real-valued model for simplicity; we will add a remark on this in our updated manuscript.
>
> > Line 99
>
> As it is standard in the optimization literature (see e.g., Bubeck (2015)), we cannot hope for truly polynomial dependence on $1/\epsilon$ unless the objective enjoys some special structure such as strong convexity. To this end, we settle for the natural $\mathrm{poly}(1/\epsilon)$ dependence. In contrast, we stress that the parameter space should be of truly polynomial size, i.e. polynomial in $\log(1/\epsilon)$.
>
> [Bubeck, Convex optimization: Algorithms and complexity, 2015]
>
> > Line 107
>
> Each point of the hypercube corresponds to a cut in the graph. The full parametrization corresponds to assigning a single parameter to any possible cut (which are exponentially many). The description is finite due to the finite precision requirement (see also our response to the previous question regarding finite precision).
>
> > Remark 5
>
> As our wording of Remark 5 shows this is a high-level/intuitive statement. The fact that optimizing (even simple) two layer neural networks is computationally intractable indicates that end-to-end optimization of deep solution samplers that can generate samples efficiently (in polynomial-time) is most likely impossible.  For more evidence we remark that having an efficient (and “small”, i.e., with polynomial number of parameters) neural network sampler that can also be provably optimized via gradient descent in polynomial iterations would imply a polynomial-time algorithm for the problems considered in this work (e.g., for Max-Cut) which is a well-known NP-hard (even to approximate) problem.
>
> > Remark 6
>
> Let us consider the weighted Max-Cut problem, with known weights. In this case, one can take the weighted Laplacian matrix and again express the loss function in the exact same form as in the unweighted case. Then the feature mappings are exactly the same.
>
> > Are there natural combinatorial optimization problems to which the framework is not applicable? If so, which problems, and why?
>
> We kindly refer to our response to Reviewer VMa2.
>
> > It seems like the proposed method is theoretically superior to what people use in practice. Is this true? And if so, why is it not the standard method in practical settings nowadays? Do you expect it to become that?
>
> Yes, our method is theoretically superior to the vanilla objective as the additional reguralization and the fast/slow mixture generators help avoid minimizers at infinity and vanishing gradients (see Section 3 of our manuscript). Our preliminary experimental evaluation indicates that some of our theoretical insights (entropy regularization, fast/slow mixing) leads to better performance. We also kindly refer to our response to Reviewer jXn4.

---

> > ### Comment · Reviewer_m1vy · 2023-08-13
> >
> > I thank the authors for their extensive response to my review and the raised questions. Trusting that the authors will incorporate the promised changes, I am increasing my score from 4 to 6.

---

### Official Review · Reviewer_VMa2 · 2023-07-07

**Soundness:** 4 excellent
**Presentation:** 4 excellent
**Contribution:** 4 excellent
**Rating:** 7
**Confidence:** 4

**Summary:**

This paper proposes a theoretical framework for analyzing the effectiveness of deep models trained by gradient-based methods as solution generators for combinatorial problems. The authors first investigate the complete, compressed and efficiently optimizable properties of the solution generator on many combinatorial tasks. To address the challenges of minimizers at infinity and vanish gradients, the authors devise an entropy regularization and a fast/slow mixture generation scheme. Experiments demonstrate that the proposed method helps address vanishing-gradient issues and escape bad stationary points.


**Strengths:**

1. The paper gives a positive answer to the existence of complete, compressed and efficiently optimizable solution generators for combinatorial tasks, and it designs a family of such solution generators.
2. The paper addresses the challenges of minimizers at infinity and vanishing gradients by the proposed entropy regularization and fast/slow mixture generation scheme.
3. A general and solid foundational theorem (Theorem 1) is clearly presented.


**Weaknesses:**

1. The authors may want to conduct experiments on more combinatorial problems with larger scales to demonstrate the effectiveness of the proposed method.
2. The authors analyze the existence and properties of the feature mappings for the solutions and instances, but how to learn such mappings remains unexplored.
3. The authors apply MLP to combinatorial problems with a fixed number of graph nodes. However, MLPs fail to process instances with varying scales. The authors may want to consider more suitable models such as the graph neural network (GNN).

**Questions:**

1. Could you please give some examples of combinatorial problems that do not satisfy Assumption 1?
2. Could you please provide more details on the representations of input data, such as the input features and data format?

**Limitations:**

Assumption 1 might be difficult to check for a general combinatorial problem, which may limit the applicability of the theoretical results in this paper.

---

> ### Author Rebuttal · Authors · 2023-08-08
>
>
> We thank the reviewer for the very positive feedback and the provided questions.
>
> > *The authors may want to conduct experiments on more combinatorial problems with larger scales to demonstrate the effectiveness of the proposed method.*
>
> *Response:* We kindly refer to our general response about experimental evaluation on large instances for Max-Cut. As a direct future direction we aim to extend our experiments to other interesting combinatorial problems.
>
>
> > *The authors apply MLP to combinatorial problems with a fixed number of graph nodes. However, MLPs fail to process instances with varying scales. The authors may want to consider more suitable models such as the graph neural network (GNN).*
>
> *Response:* We agree with the reviewer that GNNs are more suitable for such tasks and plan to use them in future experimental evaluation of our work. Since our current work is primarily theoretical, we tried to keep our simulations as simple as possible (we do not aim to improve over SOTA methods for NCO).
>
>
> > *Could you please give some examples of combinatorial problems that do not satisfy Assumption 1?*
>
> *Response:* An interesting combinatorial problem not captured by our framework is SAT.
>
> Given a combinatorial problem, Assumption 1 essentially asks for the **design** of feature mappings for the solutions and the instances that satisfy desiderata such as boundedness and variance preservation.
>
> Max-Cut, Min-Cut, TSP and Max-$k$-CSP and other problems satisfy Assumption 1 because we managed to design appropriate (problem-specific) feature mappings (see Section 2) that satisfy the requirements of Assumption 1.
>
> There are interesting combinatorial problems for which we do not know how to design such good feature mappings. For instance, the "natural" feature mapping for the Satisfiability problem (SAT) (similar to the one we used for Max-$k$-CSPs) would require feature dimension exponential in the size of the instance (we need all possible monomials of $n$ variables and degree at most $n$) and therefore, would violate item 4 of Assumption 1.
>
>
> > *Could you please provide more details on the representations of input data, such as the input features and dataformat?*
>
> *Response:* At each iteration, the input is an instance (e.g., a graph). A potential representation of the graph is the Laplacian matrix (e.g., in Max-cut). Hence, the input features are a collection of Laplacian matrices. We note that the input features depend on the combinatorial problem in hand.
>
>
> > - *The authors analyze the existence and properties of the feature mappings for the solutions and instances, but how to learn such mappings remains unexplored.*
> >
> > - *Assumption 1 might be difficult to check for a general combinatorial problem, which may limit the applicability of the theoretical results in this paper.*
>
> *Response:* In this work, we present a general framework for establishing the first theoretical understanding regarding challenging and fundamental combinatorial problems. Assumption 1 allows for our framework to be quite general and expressive. The mildness of our Assumption is justified by the number of problems it captures (Max-Cut, TSP, and Max-$k$-CSP). We emphasize that the featured mappings for the instances and the solutions correspond to a design problem; in our work, we designed appropriate feature mappings for Max-Cut, TSP and other combinatorial problems. Hence, it is not exactly the case that Assumption 1 is hard to check; the challenge is to design good feature mappings, which currently is a problem-specific task. Designing principled ways to find these mappings is an interesting direction.

---

### Official Review · Reviewer_jXn4 · 2023-07-10

**Soundness:** 3 good
**Presentation:** 4 excellent
**Contribution:** 4 excellent
**Rating:** 7
**Confidence:** 4

**Summary:**

This paper deals with a significant question at the intersection of combinatorial optimization and continuous-based optimization: is it possible to design solution generators for combinatorial problems that are (1) expressive enough to generate approximately optimal solutions, (2) tractable so that their parameterization is only polynomial in the number of inputs, and (2) efficiently optimizable so that only a polynomial number of stochastic gradient descent (SGD) steps is required to learn a parameterization with almost optimal performance? The paper answers this question in the affirmative. It develops a general framework that can be used to describe complete, compressed and efficiently optimizable solution generators, which can be instantiated to accommodate a wide range of NP-hard problems. The framework requires feature mappings for both the instances and the solutions tat have a number of desired properties. Assuming this is the case, the authors provide a very general positive result. The paper highlights two challenges related to the optimization. First, the loss function may accept a minimizer at infinity, so that gradient descent may get stuck to sub-optimal configurations. The authors address this by introducing an entropy regularization term. Second, vanishing gradients may slow down optimization, rendering it inefficient. This is addressed by introducing a mixture of a fast and a slow solution generator: the fast component helps to reach the optimal solution while the slow component helps to keep a non-zero variance throughout the optimization process, hence avoiding the problem of vanishing gradients. The authors provide experimental evidence in favor of their entropy regularizer and the fast/slow mixing scheme.

**Strengths:**

1. The paper is extremely well-written. The authors have done an excellent job presenting in a simple yet rigorous manner the main insights behind their framework and the various design choices. Every single choice is adequately justified, which makes it very easy even for non-familiar readers to grasp the main messages of this work. I also liked the the fact that the authors took extra care to elucidate some potentially confusing aspects, e.g., with respect to the P vs. NP question.
2. The main result requires very reasonable assumptions and is general enough to encompass several standard hard problems.
3. The two proposed tricks (entropy regularization and fast/slow mixing scheme) are extremely well motivated but also justified theoretically. The ablation study at the end confirms their usefulness empirically.
4. The framework is novel and a major part of its novelty precisely stems from its generality.
5. Even though I was not able to check all math details, the part that I was able to check was free of problems.

**Weaknesses:**

1. I understand that the main focus of this work is to provide a generic framework with reasonable assumptions and strong performance guarantees. I feel that the authors have delivered on that premise. That said, readers may be left with the question: what is the true empirical potential of this framework, especially when we compare it to state of the art combinatorial neural solvers? After reading the paper, I was not really clear whether this work is mainly about introducing a generic framework, or if it was also about proving its practical value on various benchmarks.

**Questions:**

1. Is this work only intended to theoreticians or to practitioners as well? If the latter is the case, then the authors would need to provide evidence for that claim, e.g., performance of their solution generators compared to other continuous solvers. I feel that this question is not answered in the current paper.
2. On a similar note, the number of nodes n=15 used in the ablation study is small - could these methods scale to larger instances? Do the authors have any take-away messages for practitioners who may be interested in trying out such methods? Is it perhaps fair to say that the proposed framework mostly serves as an abstraction but without necessarily powerful practical implications?
3. In the discussion of P vs. NP, the authors explain that sampling from their solution generators may be computationally expensive but add that technique based on Langevin dynamics may help to circumvent this issue in practice. It would be worthwhile for the authors to see whether such techniques would be of any help, especially with large instances.

**Limitations:**

None.

---

> ### Author Rebuttal · Authors · 2023-08-08
>
>
> We would like to thank the reviewer for the positive feedback and insightful questions.
>
> > - *I understand that the main focus of this work is to provide a generic framework with reasonable assumptions and strong performance guarantees. I feel that the authors have delivered on that premise. That said, readers may be left with the question: what is the true empirical potential of this framework, especially when we compare it to state of the art combinatorial neural solvers? After reading the paper, I was not really clear whether this work is mainly about introducing a generic framework, or if it was also about proving its practical value on various benchmarks.*
> >
> >
> >- *Is this work only intended to theoreticians or to practitioners as well? If the latter is the case, then the authors would need to provide evidence for that claim, e.g., performance of their solution generators compared to other continuous solvers. I feel that this question is not answered in the current paper.*
>
> *Response:* In this work, we present a general framework for establishing **theoretical** understanding regarding challenging and fundamental combinatorial problems such as Max-Cut, TSP, and Max-$k$-CSP. Hence, our work is mainly intended for theoreticians and we leave more extensive experimental evaluation of our work as an interesting question for future work.
>
> However, as our simulations (see also the additional experiments on larger instances provided in this rebuttal) show that some of the insights we obtain as a byproduct of our theoretical proof of convergence may be of practical significance. Moreover, we would like to point out an interesting connection between our theoretical work and a prior experimental paper. The work of Kim et al. (2021) uses an entropy maximization scheme in order to generate diversified candidate solutions. This experimental heuristic is quite close to our theoretical idea for entropy regularization. In our work, entropy regularization allows us to design quasar-convex landscapes and the fast/slow mixing scheme to obtain diversification of solutions.
>
> [Kim, Park, Kim, Learning Collaborative Policies to Solve NP-hard Routing Problems, 2021]
>
> >- *On a similar note, the number of nodes $n=15$ used in the ablation study is small - could these methods scale to larger instances? Do the authors have any take-away messages for practitioners who may be interested in trying out such methods? Is it perhaps fair to say that the proposed framework mostly serves as an abstraction but without necessarily powerful practical implications?*
> >- *In the discussion of P vs. NP, the authors explain that sampling from their solution generators may be computationally expensive but add that techniques based on Langevin dynamics may help to circumvent this issue in practice. It would be worthwhile for the authors to see whether such techniques would be of any help, especially with large instances.*
>
> *Response:* We refer to our previous answer and the additional simulations in the general response. In our additional simulations, we implemented our method using approximate samplers based on Langevin dynamics to circumvent this issue.  Given that even in larger instances our method outperforms the vanilla objective we are optimistic that our theoretical insights are going to be of practical significance.

---

> > ### Comment · Reviewer_jXn4 · 2023-08-21
> > **thank you for the responses**
> >
> > I thank the authors for their detailed response. I encourage them to include these clarifications in the final paper version.

---

### Author Rebuttal · Authors · 2023-08-08

**General Response**

We thank all reviewers for taking the time to read our manuscript carefully and for providing constructive and insightful feedback. We are very encouraged by the positive comments of the reviewers on the novelty and significance of our theoretical framework for Neural Combinatorial Optimization (Reviewers jXn4, dPn4), theoretical contributions (Reviewers m1vy, dPn4) and the writing quality and the clarity of the presentation of the ideas (Reviewers jXn4, VMa2, dPn4).

We provide detailed responses to each reviewer separately. We look forward to engaging in further discussion with the reviewers, answering questions, and discussing improvements.

Before that, we start with a general comment where we provide detailed additional experimental results on large instances.


**Additional Simulations - Larger Instances/Approximate Samplers**


Our experiments in the paper (which do not need Langevin dynamics for sampling) showed that even for very small instances of Max-Cut (i.e., with 15 nodes), optimizing the vanilla objective is not sufficient and the iteration gets trapped in local optima. In contrast, our method using entropy regularization always manages to find the optimal cut. A natural question raised by the reviewers is whether this improvement is also apparent in larger graphs.

We focus on the case of random $d$-regular graphs with $n$ nodes. It is well-known that for this family of graphs, with high probability as $n \to \infty$, the size of the maximum cut satisfies $\mathrm{MaxCut}(G(n,d)) = n(d/4 + P_\star \sqrt{d/4} + o_d(\sqrt{d})) + o(n)$, where $P_\star \approx 0.7632$ is a universal constant [Dembo et al., 2017].

We aim to find a good approximation for the normalized cut-value, defined as $(\mathrm{cut\_value}/n - d/4)/\sqrt{d/4}$, which (roughly speaking) takes values in $[0,P_\star]$.

We obtain approximate random samples from the density $e^f$ using the Metropolis-Adjusted Langevin Algorithm (MALA). In particular, an approximate sample from this density is obtained by the Euler–Maruyama method for simulating the Langevin diffusion: $x_{k+1} = x_k + \tau \nabla f(x_k) + \sqrt{2\tau} \xi_k$, where $\xi_k$ is an independent Gaussian vector $\mathcal{N}(0,I)$. MALA incorporates an additional step based on the Metropolis-Hastings algorithm (see [Besag, 1994, Song and Kingma, 2021]). In our case, the score function $f$ is a simple 3-layer ReLU network.

**In our additional experiments for 3 larger random regular graphs (600 nodes) using the fast/slow mixing technique along entropy regularization we see that our method leads to improvements over the vanilla objective.** Plots of the trajectories of the vanilla and our method can be found in the figures provided in the pdf of the rebuttal. In the horizontal axis we plot the iterations and in the vertical axis we plot the normalized cut score of each method (higher is better) -- we stop the plot of the vanilla trajectory after 200 iterations because we observed that its output has fully converged and is stuck.

[Dembo, Montanari and Sen, Extremal cuts of sparse random graphs, 2017]

[Besag, Comments on “Representations of knowledge in complex systems” by U. Grenander and MI Miller, 1994]

[Song and Kingma, How to train your energy-based models, 2021]

---

### Decision · Program_Chairs · 2023-09-21

**Decision:**

Accept (oral)

**Comment:**

The reviewers have provided a thorough evaluation of the paper and have generally agreed on its strengths and potential impact. The paper presents a novel theoretical framework for analyzing the effectiveness of deep models trained by gradient-based methods as solution generators for combinatorial problems. The authors have done an excellent job in presenting their work and justifying their design choices. The paper is well-written and the main insights are presented in a simple yet rigorous manner.

The reviewers have raised some concerns and questions, mainly related to the practical implications of the framework and its scalability. However, these concerns do not undermine the theoretical contributions of the paper. The authors are encouraged to address these concerns in their final version of the paper or in future work.

Overall, the paper is technically solid and has the potential to have a high impact on the field. The theoretical contributions are significant and the paper is well-positioned to stimulate further research in this area. Therefore, I recommend accepting this paper.